# Targeting Cancer Cell Fate: Apoptosis, Autophagy, and Gold Nanoparticles in Treatment Strategies

**DOI:** 10.3390/cimb47060460

**Published:** 2025-06-14

**Authors:** Maria Anthi Kouri, Alexandra Tsaroucha, Theano-Marina Axakali, Panagiotis Varelas, Vassilis Kouloulias, Kalliopi Platoni, Efstathios P. Efstathopoulos

**Affiliations:** 1Department of Applied Medical Physics, Attikon University Hospital, Medical School, National and Kapodistrian University of Athens, 11527 Athens, Greece; mariakouri90@gmail.com (M.A.K.); polaplatoni@gmail.com (K.P.); 2Medical Physics, General Hospital GHA Korgialeneio Mpenakeio-Hellenic Red Cross, 11526 Athens, Greece; 3Laboratory of Bioethics, School of Medicine, Democritus University of Thrace, 68100 Alexandroupolis, Greece; atsarouc@med.duth.gr; 4Department of Biomedical Engineering, Radiation Physics, Materials Technology and Biomedical Imaging Laboratory, AKΤYΒA, University of West Attica, Egaleo, 12210 Athens, Greece; bme19388009@uniwa.gr; 5Hemodynamic Laboratory, General Hospital GHA Korgialeneio Mpenakeio-Hellenic Red Cross, 11526 Athens, Greece; pvarelas@outlook.com; 6Department of Clinical Radiation Oncology, Attikon University Hospital, Medical School, National and Kapodistrian University of Athens, 11528 Athens, Greece; vkouloul@med.uoa.gr

**Keywords:** apoptosis, autophagy, cancer treatment, gold nanoparticles, radiationtherapy, chemotherapy

## Abstract

At the intersection of nanotechnology and cancer biology, gold nanoparticles (AuNPs) have emerged as more than passive carriers—they are active agents capable of reshaping cellular fate. Among their most promising attributes is the potential to modulate apoptosis and autophagy, two intricately linked pathways that determine tumor response to stress, damage, and treatment. Apoptosis serves as the principal mechanism of programmed cell death, while autophagy offers a dualistic role—preserving survival under transient stress or contributing to cell death under sustained insult. Thus, understanding how these mechanisms interact—and how AuNPs influence this crosstalk—may be key to unlocking more effective oncologic therapies. This review explores the molecular interplay between apoptosis and autophagy in cancer and evaluates how AuNPs impact these pathways. By enhancing radiosensitization in radiation therapy and improving drug delivery and chemotherapeutic precision, AuNPs offer a unique strategy to circumvent resistance in aggressive or refractory tumors towards shaping their biological behavior and cellular pathways and, therefore, forming a patient-centered personalized therapeutic potential. Yet, clinical translation remains challenging. The dynamic physicochemical nature of AuNPs makes their biological behavior highly context-dependent. Combined with the complexity of apoptotic and autophagic signaling and tumor heterogeneity, this creates a triad of profound intricacy. However, within this complexity lies therapeutic opportunity. Framing AuNPs, apoptosis, and autophagy as a synergistic axis may enable mechanism-informed, adaptable, and patient-specific cancer therapies. This paradigm shift invites a more strategic integration of nanotechnology with molecular oncology, advancing the frontier of precision medicine.

## 1. Introduction

In the relentless pursuit of effective cancer therapies, the challenge remains to enhance treatment efficacy while minimizing damage to normal tissues [1,2]. Both radiotherapy and chemotherapy, two cornerstone modalities in cancer treatment, exploit the intrinsic differences in the repair capabilities and survival curves of cancer and normal cells [3,4]. Radiotherapy achieves this by delivering ionizing radiation (IR) to induce DNA damage and disrupt tumor cell proliferation, while chemotherapy relies on cytotoxic agents to interfere with cancer cell growth and survival [5,6]. Recent advancements have focused on improving the therapeutic index of these treatments, incorporating innovations such as fractionated irradiation, targeted drug delivery systems, and combination therapies [7]. Among these, gold nanoparticles (AuNPs) have emerged as promising dual-function agents, serving both as radiosensitizers to enhance radiotherapy efficacy and as carriers for chemotherapeutic drugs to improve bioavailability and reduce systemic toxicity [8,9,10]. This multifaceted role underscores their potential to revolutionize cancer treatment by optimizing both radiation- and drug-based strategies.

Yet, despite these advances, the urgency for further breakthroughs cannot be overstated. While innovations in radiotherapy and chemotherapy—along with their potential enhancement through AuNPs—have significantly improved cancer treatment, they alone are not sufficient to overcome the complexities of tumor resistance and progression. A true paradigm shift requires looking beyond these macroscopic strategies and delving into the molecular and cellular landscapes that dictate therapeutic outcomes. The intricate crosstalk of DNA damage repair, apoptosis, and autophagy ultimately determines whether a cancer cell succumbs to treatment or adapts to survive [11,12]. By unraveling these fundamental mechanisms and their crosstalk, we can identify novel vulnerabilities in tumors and design more precise, effective interventions that go beyond merely delivering cytotoxic agents—interventions that actively manipulate the cell’s intrinsic death and survival pathways for maximal therapeutic impact [13,14]. Expanding the therapeutic window remains a priority, achievable through two key approaches: sensitizing tumor cells to treatment while sparing normal tissues or protecting normal tissues without compromising tumoricidal effects. Both strategies depend on a detailed understanding of the molecular and cellular mechanisms triggered by IR and chemotherapeutic agents to identify potential therapeutic targets.

Among these cellular responses, apoptosis, or programmed cell death, is a tightly regulated process crucial for eliminating irreparably damaged or potentially harmful cells [15]. This process is primarily regulated by two pathways: the intrinsic (mitochondrial) pathway and the extrinsic (death receptor) pathway [15,16]. IR commonly activates the intrinsic pathway through p53-dependent signaling, where DNA damage triggers the phosphorylation of p53 by ATM and CHK2 [17,18]. This leads to the transcriptional upregulation of pro-apoptotic genes such as BAX and PUMA, resulting in mitochondrial membrane disruption, cytochrome c release, and subsequent caspase activation [19,20]. In addition, IR-induced reactive oxygen species (ROS) can initiate p53-independent apoptosis through membrane stress pathways, further amplifying cell death signals [21]. While most normal tissues respond to IR through DNA repair or senescence, tumor cells frequently develop resistance by overexpressing anti-apoptotic proteins like Bcl-2 or downregulating pro-apoptotic factors such as Bax [22]. Overcoming this resistance remains a major focus in cancer therapy, with strategies aimed at restoring apoptotic pathways to improve treatment efficacy [23].

Autophagy, another fundamental cellular process, involves the degradation and recycling of intracellular components to maintain cellular homeostasis [24,25]. This regulating mechanism progresses through distinct stages: initiation, nucleation, elongation, and degradation, regulated by key molecular players such as mTOR, AMPK, Beclin-1, and LC3 [26,27,28]. In the context of cancer, autophagy exhibits a dual role. On one hand, it can suppress tumorigenesis by eliminating damaged organelles and proteins, thus preventing the accumulation of cellular debris [29,30]. On the other hand, autophagy can support tumor survival by providing nutrients under metabolic stress conditions, allowing cancer cells to thrive [31,32].

Moreover, autophagy often intersects with apoptosis, sharing signaling pathways and regulatory proteins such as Bcl-2 [11,33]. This crosstalk complicates its role in cancer therapy, as autophagy can either inhibit apoptosis by sustaining cell viability or promote it under prolonged stress conditions [34,35,36]. Understanding this delicate balance between apoptosis and autophagy is crucial for developing effective cancer treatments, as these mechanisms dictate how tumor cells respond to therapeutic interventions.

AuNPs, beyond their established roles in radiosensitization and drug delivery, have demonstrated potential in modulating apoptosis and autophagy—two critical pathways that dictate cancer cell fate [37,38]. Their ability to influence these mechanisms presents an opportunity to enhance therapeutic efficacy by not only improving drug and radiation responses but also actively regulating cell death and survival pathways [39]. However, despite growing interest in the intersection of nanotechnology and cancer biology, the precise impact of AuNPs on apoptosis and autophagy remains an evolving area of study. Understanding how these nanoparticles interact with the molecular machinery governing these processes is essential for unlocking their full potential in cancer therapy.

This review aims to investigate how AuNPs influence apoptosis and autophagy in cancer treatment, examining whether these two mechanisms are interconnected and, if so, how their delicate interplay with AuNPs can be leveraged for therapeutic benefit. By analyzing existing research, we seek to determine how AuNPs modulate these cellular pathways, whether they enhance or inhibit their effects, and what this means for improving cancer treatment. Additionally, we aim to identify gaps in current knowledge and explore future directions for integrating AuNP-based strategies in a way that maximizes treatment efficacy while minimizing resistance. Understanding this intricate triptych—apoptosis, autophagy, and AuNPs—could open new avenues for more precise and effective cancer interventions.

## 2. Mechanisms of Life and Death: Apoptosis and Autophagy in Cellular Dynamics

### 2.1. Apoptosis

First described in the 1970s, apoptosis was initially viewed as a counterpart to mitosis. It is now understood as the ATP-dependent, enzyme-regulated, genetically encoded mechanism of programmed cell death, essential for maintaining cellular homeostasis by eliminating cells that are either no longer needed or potentially harmful. Apoptosis involves the proteolytic dismantling of the cytoskeleton and the endonucleolytic fragmentation of DNA, leading to highly regulated, non-inflammatory cell death. It is distinct from other regulated cell death forms: necroptosis, characterized by TNF-A activation and necrotic morphology; pyroptosis, marked by inflammasome activation and membrane rupture; and ferroptosis, a lipid peroxidation-driven, iron-dependent form of non-apoptotic death [40,41,42]. Among these, apoptosis is considered immunologically silent and non-lytic, although emerging evidence indicates that under certain conditions, it may trigger immune signaling via damage-associated molecular patterns (DAMPs) [43].

From a pathological standpoint, apoptosis is characterized by cell shrinkage, cytoskeletal breakdown primarily via caspases, loss of intercellular connections, and cytoplasmic condensation. The nucleus becomes basophilic, and chromatin undergoes pyknosis, condensing into dark-staining aggregates near the nuclear envelope. Subsequent karyorrhexis fragments the nuclear material, and the resulting apoptotic bodies—small vesicles containing organelles and nuclear fragments—are efficiently removed through efferocytosis by phagocytes. The plasma membrane remains intact throughout, distinguishing apoptosis from necrosis or pyroptosis, which are associated with cell lysis and inflammation. Macrophages typically clear apoptotic cells rapidly and silently, minimizing collateral tissue damage [43].

#### 2.1.1. Apoptotic Pathways

Apoptosis proceeds via two primary signaling routes: the intrinsic (mitochondrial) and extrinsic (death receptor) pathways. The intrinsic pathway is activated in response to intracellular stress, such as DNA damage from radiation or chemotherapeutic agents, hypoxia, or the accumulation of misfolded proteins—as observed in neurodegenerative diseases like Alzheimer’s, Parkinson’s, and Huntington’s diseases. These insults lead to mitochondrial outer membrane permeabilization and cytochrome c release into the cytosol, where it binds apoptotic protease activating factor-1 (APAF-1), forming the apoptosome and activating caspase-9. This process is tightly regulated by members of the Bcl-2 family and the TP53 tumor suppressor gene, which modulates the expression of pro-apoptotic proteins like Bax and PUMA [44,45].

The extrinsic pathway is triggered by external signals binding to death receptors on the cell surface. TNF-A, produced by macrophages, binds TNFR1 to initiate caspase activation. Similarly, T lymphocytes expressing Fas ligands bind to Fas receptors on target cells, promoting apoptosis to downregulate immune responses post-infection. Fas, a TNF family member, plays a central role in immune homeostasis. Bcl-2, located on chromosome 18, exerts anti-apoptotic effects by binding APAF-1 and preventing cytochrome c release from mitochondria [44]. TP53, activated by DNA damage, induces G1 phase arrest and promotes apoptosis by upregulating Bax, which in turn neutralizes Bcl-2. CD8+ cytotoxic T cells contribute to extrinsic apoptosis via perforin and granzyme release—perforins form membrane pores, allowing granzymes to enter and activate caspases [45].

Caspases, cysteine–aspartic proteases, are key effectors in both pathways. They are synthesized as inactive proenzymes and require cleavage for activation. Initiator caspases [41,46,47,48] activate effector caspases [42,45,49], which then orchestrate the degradation of cellular components. Caspase-3, a principal executioner, catalyzes chromatin condensation and cleaves structural and repair proteins. It also activates DNases, leading to internucleosomal DNA fragmentation [45,46,49]. Recent research reveals non-apoptotic roles for caspases in differentiation, inflammation, and stem cell regulation, broadening their functional relevance in cancer biology [46].

#### 2.1.2. Role of Apoptosis in Maintaining Tissue Homeostasis

Apoptosis plays an essential role in morphogenesis and tissue remodeling during development. For instance, digit formation involves the selective elimination of interdigital cells through apoptosis, and male fetuses lose Müllerian structures via Sertoli cell-derived Müllerian inhibitory factor [47,48]. In adults, endometrial shedding during menstruation occurs via hormone-regulated apoptosis [50]. The removal of misfolded proteins, including prion and amyloid aggregates, also depends on apoptotic pathways, potentially preventing neurodegenerative conditions.

In the immune system, apoptosis ensures both central and peripheral tolerance. Autoreactive T and B lymphocytes are eliminated during development, preventing autoimmunity. Following infections, neutrophils and activated lymphocytes are removed by apoptosis to resolve inflammation. Corticosteroids promote lymphocyte apoptosis, contributing to their immunosuppressive action. Cytotoxic CD8+ T cells use apoptotic mechanisms to eliminate infected or malignant cells. Radiation and chemotherapeutic agents trigger DNA damage and p53-mediated cell cycle arrest at the G1 phase; if repair is unsuccessful, p53 initiates apoptosis. A defective p53 pathway allows abnormal cells to survive, promoting tumorigenesis. In the thymus, autoreactive T cells are deleted via apoptosis to preserve self-tolerance [51].

Beyond its role in eliminating damaged or superfluous cells, apoptosis actively contributes to tissue regeneration. Dying cells can release mitogenic factors—such as Wnt proteins and prostaglandin E2 (PGE2)—which stimulate neighboring progenitor and stem cells to proliferate, a process known as apoptosis-induced proliferation (AiP) [52]. This has been demonstrated in several models, including regenerating skin and liver, as well as in Drosophila, where caspase activation in apoptotic cells drives compensatory proliferation [52].

In mammals, apoptotic cell clearance via efferocytosis further promotes regeneration by dampening inflammation and fostering a reparative microenvironment. Recent single-cell RNA sequencing in murine skin wounds revealed that apoptotic signaling and efferocytosis receptors, including Timd4 and Axl, are upregulated in fibroblasts and macrophages during early inflammation [53]. Functionally, Timd4 was shown to be essential for both apoptotic clearance and tissue repair, while Axl promoted regeneration through separate, efferocytosis-independent pathways [53].

In chronic wounds such as diabetic ulcers, dysregulation of pathways like Axl/Gas6 impairs healing, highlighting the clinical relevance of apoptotic signaling in repair processes [53]. Across species—from Hydra to mammals—AiP is a conserved mechanism that mobilizes quiescent stem cells in response to injury, with caspase activity triggering regenerative proliferation [44,54,55,56,57,58,59].

In therapeutic settings, mesenchymal stromal cells (MSCs) administered for tissue repair often undergo apoptosis shortly after infusion. Their apoptotic clearance by monocytes reprograms immune responses toward anti-inflammatory, pro-regenerative phenotypes—supporting repair not by cell replacement but by immune modulation [60].

However, the regenerative benefits of apoptotic signaling may be co-opted in cancer. Apoptosis induced by chemo- or radiotherapy can inadvertently stimulate surviving malignant cells through mitogenic cues, potentially facilitating tumor repopulation and therapy resistance [61]. Thus, apoptosis serves as both a regulator of tissue renewal and a potential driver of tumor progression, emphasizing the importance of context in its therapeutic targeting.

#### 2.1.3. Apoptosis in Cancer

The progression from normal to malignant phenotype involves evasion of apoptosis, enabling cell survival despite oncogenic stress and genomic instability [62]. Physiological signals that typically induce apoptosis—such as DNA damage or oncogene activation—are suppressed in cancer cells. The tumor suppressor p53 is central to this regulation: upon detecting genomic insults, it activates pro-apoptotic proteins like PUMA, NOXA, and BAX [63]. Cytoplasmic p53 also neutralizes Bcl-2 and Bcl-xL, removing inhibitory control over Bax/Bak and promoting mitochondrial permeabilization [64].

Oncogenes like MYC can paradoxically promote apoptosis under certain conditions, primarily through BIM-dependent pathways. However, cancer cells often bypass this response by upregulating anti-apoptotic Bcl-2 family members or downregulating pro-apoptotic effectors [65]. This imbalance allows cancer cells to evade programmed cell death and resist treatment-induced cytotoxicity [66].

Targeting apoptosis pathways has become a key therapeutic focus. BH3 mimetics, such as venetoclax, are designed to inhibit Bcl-2 and restore apoptotic potential in tumor cells, with promising results in hematological malignancies. By reinstating the apoptotic threshold, such therapies aim to sensitize tumors to chemoradiation and reduce resistance. Furthermore, recent insights into the immunogenic potential of apoptosis challenge the long-held notion of its immunological silence, revealing that in some contexts, apoptotic cells may modulate immune surveillance via DAMP release [43].

### 2.2. Autophagy

Cells are continually faced with a variety of internal and external stressors that necessitate effective mechanisms to maintain balance and prevent cell death. Autophagy, particularly macroautophagy, is a crucial process responsible for the breakdown and recycling of damaged organelles and misfolded proteins [67,68]. It also plays a role in responding to diverse stimuli that can impact cellular health. Eukaryotic cells depend on autophagy for their housekeeping functions, which help to regulate cellular homeostasis [68,69]. As a result, autophagy generally operates at low levels within cells but can be upregulated under stress conditions such as oxidative or metabolic stress [68,70]. This highly conserved process enables cells to degrade and recycle various components, thereby promoting overall cellular health [71].

The autophagy process begins with the formation of double-membrane vesicles called autophagosomes, which encapsulate cellular debris, including damaged organelles and proteins [57,71]. These autophagosomes subsequently fuse with lysosomes, where the contents are broken down and recycled into metabolites that the cell can use as building blocks for biosynthesis [68,72]. Autophagy is essential for normal cell function, and its dysregulation has been linked to the development of various diseases, including neurological disorders, diabetes, cardiovascular conditions, and particularly cancer [69,73,74]. Therefore, targeting and modulating autophagy represents a significant avenue for therapeutic interventions.

#### 2.2.1. Autophagy Pathways

In mammalian cells, three primary types of autophagy exist: macroautophagy, microautophagy, and chaperone-mediated autophagy [75]. The autophagic process involves a series of molecular mechanisms regulated by various proteins, making their identification crucial for developing targeted therapies aimed at modulating autophagy [76,77].

Different cellular stressors, such as nutrient deprivation and metabolic stress, can trigger autophagy activation through the phosphorylation and subsequent activation of 5′-AMP-activated protein kinase (AMPK), primarily mediated by upstream kinases such as liver kinase B1 (LKB1) and calcium/calmodulin-dependent protein kinase kinase 2 (CAMKK2) [78,79,80]. Activated AMPK, in turn, promotes autophagy by inhibiting the mTOR pathway and activating the ULK1 complex, which is composed of ULK1, FIP200, ATG13, and ATG101. The ULK1 complex plays a crucial role in autophagy initiation by phosphorylating Beclin-1 at Ser15 and Ser30, as well as ATG14 at Ser29, particularly under conditions of energy stress or mTORC1 [81,82,83].

The subsequent formation of autophagosomes involves the lipidation of LC3-I (light chain 3-I) to produce LC3-II, a process that requires the action of proteins such as ATG5 and ATG16L1 [84,85,86]. Following the formation of the autophagosome, it fuses with a lysosome, a process facilitated by proteins such as LAMP2 and SNAREs [86]. This fusion allows the degradation of the autophagosomal cargo, which involves various enzymes, including proteases, lipases, and nucleases [86]. As illustrated in Figure 1, this process is exemplified by the selective degradation of a damaged mitochondrion, representing mitophagy as a specific form of autophagy [77].

Autophagy, which is often referred to as self-digestion, is a well-conserved process that plays an essential role in maintaining cellular health, differentiation, and overall homeostasis [58,78,79]. It typically involves two stages: the encapsulation of cytoplasmic components into double-membraned vesicles known as autophagosomes, followed by their fusion with lysosomes to form autolysosomes, where degradation occurs [61,80]. According to the Guidelines for the use and interpretation of assays for monitoring autophagy (4th edition) by Klionsky et al., autophagosomes are typically double-membraned; although multilamellar membrane structures can occasionally be observed, these are not characteristic and may reflect overlapping phagophores, autolysosomes, or artifacts of sample preparation [87]. Thus, the role of autophagy is complex, as it can either support cell survival or lead to cell death depending on the context and level of autophagy activation [24]. This process can be activated in response to various stimuli, including nutrient deprivation, infections, organelle damage, DNA damage, and treatment with radiotherapy or chemotherapy [12,13,88,89].

#### 2.2.2. Autophagy in Cancer

Autophagy plays a complex role in cancer, acting as both a tumor suppressor and a promoter, depending on the context [90]. In its protective role, autophagy maintains cellular homeostasis by clearing damaged organelles, reducing oxidative stress, and preventing genomic instability—key factors in tumor suppression [24,29,91]. Dysregulated autophagy has been linked to increased DNA damage, chronic inflammation, and cancer initiation [13,92,93].

Conversely, in established tumors, autophagy often supports cancer cell survival under metabolic stress, hypoxia, or nutrient deprivation by recycling cellular components to sustain energy production [27,31,94]. This adaptive mechanism can also promote treatment resistance and tumor recurrence by enabling dormant cancer cells to survive harsh conditions [95,96].

Key autophagy regulators, such as Beclin-1, ATG5, and ATG7, are often altered in various cancer types [97]. Beclin-1 downregulation is linked to breast, ovarian, and prostate cancers, while heightened autophagic activity in advanced pancreatic and colorectal cancers is associated with enhanced survival and therapy resistance [98,99]. Furthermore, recent research has begun to uncover the impact of epigenetic modifications—such as DNA methylation, histone modifications, and miRNA regulation—on the regulation of autophagy [100,101,102]. Aberrant expression of autophagic genes can either enhance or suppress autophagy, which in turn can significantly affect cancer progression and treatment resistance [26,31,58,103].

### 2.3. Interplay Between Apoptosis and Autophagy

As has been previously described, autophagy and apoptosis constitute two fundamental processes governing cell fate, both of which are intricately linked to cancer progression and therapeutic response. Their interaction exhibits a dual role, with autophagy either facilitating apoptotic cell death or promoting tumor cell survival, depending on the cellular context [43]. As illustrated in Figure 2, these pathways are closely interconnected through both direct molecular interactions and context-dependent signaling feedback loops, with each process capable of regulating or suppressing the other under specific physiological or pathological conditions.

#### 2.3.1. Regulation of Apoptosis on Autophagy

Apoptosis reciprocally regulates autophagy through the caspase-mediated degradation of autophagy proteins [104]. Caspase cleavage of Atg3 inhibits autophagosome formation, while combined caspase and calpain activity degrades AMBRA1, suppressing autophagic initiation [105]. Additionally, caspase-mediated cleavage of Beclin-1 results in the formation of a C-terminal fragment that enhances apoptosis but disrupts autophagic flux [106].

During mammalian development, apoptosis indirectly promotes autophagy by generating apoptotic debris, which autophagy clears to prevent secondary necrosis [15,107]. In tumors, impaired clearance of apoptotic bodies due to defective autophagy can induce oncogenic mutations, promoting tumor survival [104,108,109]. Conversely, autophagy can counteract genomic instability under metabolic stress by eliminating damaged mitochondria, reducing ROS accumulation, and suppressing DNA damage responses [110,111,112].

The regulatory crosstalk between autophagy and apoptosis is context-dependent and involves direct molecular interactions. Thereinafter, disrupting this balance contributes to consequent pathological conditions, including neurodegeneration and tumor progression [113,114].

#### 2.3.2. Regulation of Autophagy in Apoptosis

Autophagy serves as both a pro-survival and pro-death mechanism in apoptosis regulation [115]. Under oxidative or nutrient stress, autophagy preserves cellular homeostasis by degrading damaged mitochondria and suppressing apoptotic signaling [112,116]. It prevents apoptosis by eliminating pro-apoptotic factors, such as caspase-8, and maintaining mitochondrial integrity [117,118]. However, excessive autophagy can lead to apoptosis, particularly in tumor cells with defective apoptotic pathways [24]. It has been observed that autophagy-related proteins such as Atg5 and Atg12 facilitate apoptosis by interacting with the mitochondrial apoptotic machinery, promoting caspase activation [35,119]. More precisely, Yousefi et al. demonstrated that Atg5 undergoes calpain-mediated cleavage, generating a pro-apoptotic fragment that localizes to mitochondria, leading to cytochrome c release and caspase activation [120]. Furthermore, autophagy contributes to apoptosis by degrading anti-apoptotic proteins, such as Bcl-2, disrupting mitochondrial membrane integrity and enabling apoptosome formation [11]. In some cases, autophagosomes serve as platforms for apoptotic signaling complexes, amplifying apoptosis [121].

A summary of the aforementioned key interactions between autophagy and apoptosis is illustrated in Figure 2.

#### 2.3.3. Interplay of Apoptosis and Autophagy in Cancer

The research of Chen et al. demonstrated that lysosome-mediated autophagy and caspase-dependent apoptosis play essential roles in maintaining cellular homeostasis, particularly in the regulation of tumor progression and response to therapy. They highlighted the role of high-mobility group box 1 (HMGB1) as a key modulator of autophagy and apoptosis [122,123,124]. The subcellular localization, oxidative state, and binding interactions of HMGB1 determine the dominance of either autophagy or apoptosis, thereby influencing tumor progression and therapeutic resistance [124,125]. Their findings suggest that targeting HMGB1 could serve as a potential strategy to modulate the balance between autophagy and apoptosis in cancer therapy [124].

The interaction between apoptosis and autophagy is particularly evident in glioma, one of the most aggressive brain tumors [126]. The research of Fan et al. indicated that simultaneous induction of apoptosis and pro-death autophagy significantly decreases glioma cell survival, providing a potential therapeutic avenue for glioblastoma treatment [126]. Additionally, it has been reported that rapamycin enhances miRNA-26-5p expression, leading to the downregulation of death-associated protein kinase 1 (DAPK1), a crucial regulator of autophagy. This downregulation promotes autophagy-mediated apoptosis, thereby reducing glioma cell proliferation and viability [127].

Conversely, autophagy can act as a pro-survival mechanism, allowing cancer cells to evade apoptosis and resist treatment [27]. In breast cancer, the study of Du et al. revealed that matrine, an anti-tumor agent, induces apoptosis; however, it concurrently activates cytoprotective autophagy, which counteracts its cytotoxic effects [128]. This suggests that inhibiting autophagy in combination with matrine treatment could enhance its efficacy by preventing tumor cells from escaping apoptosis [129]. Further studies on breast cancer confirmed the role of autophagy in modulating apoptosis, underscoring the necessity of context-dependent therapeutic approaches [130,131,132].

In lung cancer, the research of Lei et al. demonstrated that the downregulation of the Akt and Hedgehog signaling pathways by Jervine induces autophagy-dependent apoptosis, highlighting a novel therapeutic strategy. Similarly, in bladder cancer, research showed that therapy response is influenced by the interaction between autophagy and apoptosis [133,134,135]. Their findings revealed that ROS overgeneration induced by artenusate leads to autophagy-mediated apoptosis, effectively restricting bladder cancer progression [134,135].

The complex interplay of autophagy and apoptosis is also observed in prostate cancer, where their regulation affects tumor cell proliferation and therapy resistance [136,137]. Multiple studies have demonstrated that manipulating autophagic pathways can either potentiate apoptosis or enable cancer cells to survive under therapeutic pressure [108,138]. The ability of autophagy to promote cancer cell survival in hypoxic tumor regions has been linked to radioresistance, emphasizing the need for autophagy modulation in radiotherapy [139,140,141,142,143].

## 3. Gold Nanoparticles in Cancer Treatment

AuNPs exhibit unique physical, chemical, and biological properties that make them ideal candidates for cancer therapy [10,144,145,146]. Their small size (ranging from 1 to 100 nm) allows for efficient cellular uptake and deep tumor penetration, while their polymorphic nature enables customization of their shape (spherical, rod-shaped, or star-shaped) for specific biomedical applications [147,148]. Chemically, AuNPs are highly stable, resistant to oxidation, and capable of functionalization with biomolecules or polymers, such as polyethylene glycol (PEG), to enhance their stability and biocompatibility in biological environments [149,150]. Biologically, AuNPs exhibit low toxicity, high biocompatibility, and the ability to accumulate in tumor tissues due to the enhanced permeability and retention (EPR) effect, making them suitable for targeted drug delivery and radiosensitization [8,146].

### 3.1. Gold Nanoparticles as Radiosensitizers

Gold nanoparticles act as radiosensitizers due to their high atomic number and electron density, which enhance energy deposition upon IR exposure. This leads to increased ROS generation and exacerbated DNA damage within tumor cells. AuNPs can also modulate cellular pathways, influencing apoptosis and autophagy. Studies indicate that AuNPs promote mitochondrial membrane disruption, leading to cytochrome c release and caspase activation, thus facilitating apoptosis [151,152,153]. Concurrently, AuNPs interfere with lysosomal function or directly activate autophagic pathways, potentially inducing autophagic cell death in apoptosis-resistant cancer cells [154,155].

By selectively amplifying IR-induced cytotoxicity in tumor cells while sparing normal tissues, AuNPs expand the therapeutic window of radiotherapy. Their modulation of programmed cell death pathways offers a strategy to overcome therapeutic resistance and improve clinical outcomes. Additionally, their radiosensitization effects are quantified through the dose enhancement factor (DEF) and sensitization enhancement ratio (SER), which are critical parameters in radiotherapy efficacy [156,157,158].

Although AuNPs show significant radiosensitization at kilovoltage (kV) energy levels, extending their application to megavoltage (MeV) radiation remains a challenge due to reduced dose enhancement at higher photon energies [151]. The most promising results are observed in the kV range, where the photoelectric effect dominates, significantly enhancing radiation absorption. However, clinical radiotherapy primarily operates in the MeV range, posing a challenge for AuNP-mediated dose enhancement [159,160]. Encouragingly, emerging research suggests that AuNPs can still provide beneficial radiosensitization effects even at MeV energies [161]. While the dose enhancement factors (DEF) in this range are lower than in the kV range, even modest enhancements remain clinically significant, as any increase in radiation efficacy can contribute to improved therapeutic outcomes [162]. Moreover, recent studies indicate that AuNPs can enhance apoptosis in the MeV range by modulating key cell death pathways, further supporting their potential in radiotherapy optimization [163].

#### Optimal Radiosensitization with Gold Nanoparticles

For effective radiosensitization, optimizing AuNP cellular uptake is crucial, as increased intracellular gold concentrations enable reduced radiation doses while maintaining therapeutic efficacy. Several studies have evaluated the influence of AuNP size, shape, and surface chemistry on cellular internalization. Penninckx et al. demonstrated that spherical AuNPs of 50 nm exhibited the highest uptake compared to 14, 30, 74, and 100 nm particles, correlating with enhanced radiosensitization [151]. Similarly, spherical AuNPs outperformed rod-shaped and star-shaped variants in cellular internalization and sensitization efficiency [151,159,164].

Surface modifications further enhance AuNP uptake and therapeutic outcomes. Citrate-stabilized AuNPs demonstrate superior uptake via serum protein-mediated endocytosis compared to transferrin-coated AuNPs [144]. Cationic modifications, such as 16-mercaptohexadecyl trimethylammonium bromide (MTAB) and polyethylenimine, improve uptake through electrostatic interactions with negatively charged cell membranes [165]. Targeting ligands like arginine–glycine–aspartate (RGD) peptides have been utilized to direct AuNPs toward integrins overexpressed in tumor vasculature, significantly enhancing radiosensitization [166,167].

Additionally, protein-capped AuNPs, such as bovine serum albumin (BSA) and glutathione-protected nanoclusters, exhibit prolonged systemic circulation and enhanced tumor accumulation via the enhanced permeability and retention (EPR) effect [168]. These findings emphasize the importance of precise engineering of AuNPs in terms of size, shape, and surface chemistry to maximize their radiosensitization potential in cancer therapy [146].

Of course, it is important to acknowledge that gold nanoparticles are not the only nanomaterials with radiosensitizing capabilities. Several other platforms have shown promise in enhancing radiotherapy by modulating apoptosis and autophagy beyond their role in eliminating damaged ions. For instance, titanium dioxide (TiO_2_) nanoparticles can generate reactive oxygen species under ionizing or UV-A radiation, triggering oxidative stress and apoptotic cell death [169]. Similarly, iron oxide nanoparticles (IONPs) have been found to enhance radiosensitivity by promoting ferroptosis and autophagy-related mechanisms, particularly in challenging tumor microenvironments [170,171,172,173]. A clinically advanced example is hafnium oxide (HfO_2_) nanoparticles, such as NBTXR3, which increase radiation dose deposition and induce apoptosis in solid tumors through their high atomic number and interaction with therapeutic radiation [170,172,174]. That said, gold nanoparticles still offer distinct advantages that support their continued prominence in this field. Their high degree of surface tunability, well-documented biocompatibility, and dual ability to modulate apoptotic and autophagic pathways make them uniquely versatile. While other nanomaterials are promising in specific contexts, AuNPs remain among the most adaptable and extensively studied radiosensitizers with translational potential [151].

### 3.2. Gold Nanoparticles as Drug Carriers

AuNPs have also emerged as highly efficient drug carriers due to their tunable size, shape, and surface chemistry, which allow for precise control over drug loading and release profiles [9,175]. Their small size enables deep tumor penetration and effective cellular internalization, while their large surface-to-volume ratio facilitates high drug-loading capacity [176]. Functionalization with biocompatible polymers such as polyethylene glycol (PEG) enhances circulation time and reduces immunogenicity, allowing for improved pharmacokinetics and biodistribution [177].

The surface of AuNPs can be modified with targeting ligands, such as antibodies, peptides, or aptamers, to enable selective drug delivery to cancer cells while sparing healthy tissues [144,178]. Additionally, stimuli-responsive AuNPs have been engineered to release therapeutic payloads in response to pH, temperature, or enzymatic activity, ensuring controlled drug release within the tumor microenvironment [179,180].

#### Optimal Drug Delivery with Gold Nanoparticles

To achieve optimal drug delivery, AuNPs must efficiently encapsulate and release therapeutic agents while maintaining stability in biological systems. Various factors, such as particle size, charge, and functionalization, influence their drug delivery efficiency. Studies indicate that AuNPs between 10 and 100 nm exhibit superior cellular uptake and prolonged systemic circulation, enhancing drug accumulation within tumors [144,181,182]. Surface charge modifications can further regulate interactions with cell membranes, with cationic AuNPs demonstrating enhanced uptake via electrostatic attraction to negatively charged cancer cell membranes [183].

Furthermore, AuNPs can serve as carriers for chemotherapeutic agents such as doxorubicin, paclitaxel, and cisplatin, improving their solubility, stability, and therapeutic index [184,185]. Research has shown that conjugating AuNPs with nucleic acid-based drugs, such as small interfering RNA (siRNA) or microRNA (miRNA), enhances gene silencing efficiency, offering a promising strategy for cancer gene therapy [186,187].

## 4. Gold Nanoparticles in Regulating Autophagy and Apoptosis for Cancer Treatment

### 4.1. Gold Nanoparticles in Apoptosis

#### 4.1.1. Gold Nanoparticles as Radiosensitizers in Apoptosis Induction

Upon internalization via endocytosis, gold nanoparticles accumulate in cellular organelles such as mitochondria, lysosomes, and the endoplasmic reticulum [188,189]. Once localized, AuNPs promote the generation of ROS, including hydroxyl radicals, which induce oxidative stress [190]. This oxidative stress causes extensive damage to cellular components, compromising DNA integrity and disrupting organelle function. The heightened ROS levels are particularly significant in enhancing radiosensitization, as they amplify the DNA damage induced by ionizing radiation [191,192,193]. However, beyond DNA, ROS-mediated damage also affects mitochondria by promoting mitochondrial membrane permeabilization, cytochrome c release, and, therefore, caspase activation, which constitute key apoptotic events [194,195]. Additionally, ROS can impair lysosomal degradation and disrupt ER homeostasis, collectively intensifying cellular stress. Eventually, this multi-level damage amplifies the apoptotic response, reinforcing the role of AuNPs as effective radiosensitizers that enhance radiotherapy outcomes by promoting tumor cell destruction [152,156].

Bemidinezhad et al. investigated the radiosensitizing potential of glucose-coated gold nanoparticles (Glu-GNPs) and liposome-encapsulated gold ions (Gold-Lips) in enhancing the radiation sensitivity of B16F0 melanoma cells [196]. The research team synthesized GNPs, Glu-GNPs, and Gold-Lips and characterized their physicochemical properties via dynamic light scattering (DLS) [196]. Cell viability and radiation sensitivity were assessed through MTT and colony formation assays, while apoptosis induction was evaluated by flow cytometry [196]. Additionally, intracellular ROS levels and mRNA expression of apoptosis-related genes, including Bax, Bcl-2, p53, Caspase-3, and Caspase-7, were analyzed [196]. The study revealed that both Gold-Lips and Glu-GNPs enhanced radiosensitivity in melanoma cells, with Gold-Lips displaying superior effects. Gold-Lips treatment significantly upregulated Bax, p53, and Caspase-3/-7 while downregulating Bcl-2, confirming their ability to induce apoptosis and augment radiation therapy efficacy [196,197].

In a separate study, Tsai et al. explored the role of AuNPs as radiosensitizers in enhancing the effects of Cs-137 γ-ray irradiation in human epidermoid carcinoma (A431) cells [198]. Their investigation demonstrated that 55 nm AuNPs, when combined with γ-ray exposure, significantly enhanced ROS accumulation, leading to mitochondrial dysfunction and cytoskeletal disruption [197]. Laser scanning confocal microscopy (LSCM) confirmed enhanced ROS production, while clonogenic assays revealed a radiosensitization enhancement factor of 1.29 at a 30% survival fraction [198]. This study provided clear evidence that GNPs amplified the tumoricidal effects of radiation therapy by promoting oxidative stress and apoptosis [8,151,198].

In a multimodal therapeutic approach, another research team engineered a theranostic gold-magnetic core–shell nanostructure functionalized with folate for enhanced targeting of HPV-positive oropharyngeal cancer [199]. In vitro experiments demonstrated that while gold nanoparticles, radiotherapy, or electric field application alone exhibited minimal cytotoxicity, the combination of these modalities significantly enhanced apoptosis [156,199]. This synergistic outcome highlights the potential of AuNP-based combination therapies in overcoming cancer cell resistance to radiation [199,200].

Taheri et al. designed gold nanoparticle-containing niosomes (Nio-AuNPs) and investigated their combined effect with X-ray radiation therapy (XRT) in A549 lung cancer cells [201]. The results showed that Nio-AuNPs significantly enhanced XRT-induced apoptosis. Flow cytometry and MTT assays confirmed the synergistic effects of Nio-AuNPs and XRT, further supporting their potential as effective radiosensitizers [201,202].

In another study, Saberi et al. evaluated the effect of AuNPs on HT-29 colorectal cancer cells in the context of 9 MV radiation [203]. While AuNPs alone showed minimal cytotoxicity, their combination with radiation significantly increased apoptosis rates, as confirmed by flow cytometry and Annexin V-FITC/propidium iodide staining [203,204]. The results suggest that AuNPs effectively promote radiation-induced apoptosis through enhanced DNA damage and mitochondrial dysfunction [203,205].

Exploring combined therapy approaches, Neshastehriz et al. investigated gold-coated iron oxide nanoparticles (Au@IONPs) in the context of thermo-radiotherapy [206]. U87-MG glioma cells were treated with Au@IONPs in combination with hyperthermia (43 °C for 1 h) and X-ray radiation (2 and 4 Gy) [206,207]. The combined treatment substantially enhanced apoptosis via ROS-mediated damage and mitochondrial stress, reinforcing the effectiveness of Au@IONPs as thermo-radio-sensitizers [206,208,209].

Furthermore, Hu et al. took a novel approach by developing a gold nanoparticle-based probe (AuNP-pep@Mem) that detects apoptosis through caspase-3 activity [210]. This nanoprobe allows real-time fluorescence tracking of apoptotic events in living cells, offering a promising strategy for monitoring cell death in cancer therapy [210,211,212].

#### 4.1.2. Gold Nanoparticles as Drug Carriers in Apoptosis Induction

Gold nanoparticles have also emerged as versatile platforms for the targeted delivery of therapeutic agents, including proteins, peptides, oligonucleotides, and chemotherapeutic drugs [213]. Their distinctive physicochemical properties—such as high surface area, tunable size, and facile surface functionalization—enable improved drug solubility, cellular uptake, and bioavailability [214]. Thus, chemotherapeutic agents like paclitaxel, cisplatin, and doxorubicin have been successfully conjugated to AuNPs, enhancing their therapeutic index and simultaneously overcoming drug resistance [215]. These conjugates are often linked via pH-sensitive, redox-responsive, or light-activated linkers, ensuring controlled drug release at the tumor site [216,217]. Upon internalization, AuNP–drug complexes accumulate in lysosomes or other cellular compartments, where the therapeutic agents are released to exert their cytotoxic effects. Beyond their role as passive carriers, AuNPs themselves can induce apoptosis through the modulation of signaling pathways and ROS generation [218]. This dual capability—combining precise drug delivery with intrinsic apoptotic potential—underscores their value in enhancing chemotherapy outcomes, particularly in resistant tumor models [38,219].

Surapaneni et al. examined the role of surface charge in the apoptotic effects of AuNPs in triple-negative breast cancer (TNBC) cells [220]. Their findings demonstrated that negatively charged citrate-capped AuNPs increased histone deacetylation and induced apoptosis, while positively charged cysteamine-capped AuNPs activated the p38 MAPK signaling pathway [220]. This study highlighted the importance of AuNP surface properties in modulating apoptotic pathways in cancer cells [220,221].

Gold nanopeanuts (AuP NPs) have demonstrated potent anticancer effects by activating the c-Jun N-terminal kinase (JNK) signaling pathway, a crucial component of the MAPK superfamily [222,223]. This pathway is triggered by oxidative stress caused by intracellular glutathione depletion and subsequent ROS/RNS generation [224]. These oxidative changes initiate JNK activation, which is strongly linked to apoptosis induction [225]. To confirm JNK’s role in AuP NP-mediated apoptosis, SKOV-3 ovarian cancer cells were pretreated with the JNK inhibitor II (SP600125) prior to AuP NP exposure [226]. Pretreatment significantly reduced AuP NP-induced cytotoxicity, marked by increased cell viability and decreased caspase-positive cells, confirming JNK’s pivotal role in apoptosis induction [227,228]. Further analysis showed that JNK activation was essential for downstream caspase activation, reinforcing its role as a key mediator in AuP NP-induced cell death. By targeting the ROS/JNK axis, AuP NPs present a promising strategy to enhance apoptosis in treatment-resistant cancers [229,230].

Maddah et al. investigated the apoptotic effects of AuNPs in HCT-116 colon cancer cells, revealing their potential as effective pro-apoptotic agents [231]. Treatment with AuNPs (25 and 50 µg/mL) for 48 h significantly upregulated pro-apoptotic genes, including Bax and p53, while downregulating the anti-apoptotic gene Bcl-2 [231]. Flow cytometry analysis confirmed a dose-dependent increase in apoptotic cell populations. Hoechst 33,258 nuclear staining further demonstrated chromatin condensation and nuclear fragmentation, confirming apoptosis induction [231,232]. These findings suggest that AuNPs effectively modulate apoptotic pathways in colon cancer cells, offering potential as therapeutic agents [231,233].

Radaic et al. investigated phosphatidylserine-capped gold nanoparticles (PS-AuNPs) and their apoptotic effects in breast and prostate cancer cells [234]. PS-AuNPs (50 µg/mL) significantly induced apoptosis through enhanced histone fragmentation, morphological changes, and elevated caspase-3 activity [234]. Light microscopy images showed distinctive apoptotic features, such as cell shrinkage and membrane blebbing, in PS-AuNP-treated cancer cells, while healthy cells remained unaffected [235]. These results highlight the potential of PS-AuNPs as selective apoptotic inducers, presenting a novel therapeutic strategy for targeting breast and prostate cancers, as can be depicted in Figure 3 [234].

Daei et al. explored the apoptotic and anti-angiogenic effects of AuNPs in human bladder cancer (5637) cells [236]. The study demonstrated that AuNPs at concentrations of 25 and 50 µg/mL significantly reduced cell viability in a dose-dependent manner [236]. AuNP treatment induced ROS production, upregulated Bax, and downregulated Bcl-2 and VEGFA, indicating effective apoptotic pathway modulation [236]. Flow cytometry analysis further confirmed apoptosis induction, while a wound healing assay showed reduced cell migration [236,237].

Another recent study examined the therapeutic potential of gold nanoparticles conjugated with doxorubicin (DOX) using pH-sensitive and pH-resistant linkers in oral squamous cell carcinoma [238]. Cytotoxicity, cellular uptake, and nuclear accumulation were evaluated, showing that pH-resistant DOX-AuNPs improved nuclear drug localization, resulting in a two-fold increase in apoptosis compared to pH-sensitive conjugates [238]. In vivo experiments demonstrated superior tumor shrinkage and improved survival rates in animals treated with pH-resistant DOX-AuNPs without adverse effects on blood cell counts, as illustrated in Figure 4 [238,239]. These findings suggest that AuNPs enhance the cytotoxic effects of DOX by promoting nuclear drug accumulation and apoptosis in cancer cells [238].

Additionally, pemetrexed-conjugated AuNPs (PEM-AuNPs) were developed to induce apoptosis in NSCLC cells. PEM-AuNPs at concentrations of 50, 100, and 200 µM significantly reduced cell viability in A549 and H1299 cells in a dose-dependent manner [240]. Acridine orange–ethidium bromide and DAPI staining revealed nuclear condensation and DNA fragmentation in treated cells. Additionally, DCFH-DA staining confirmed increased ROS generation, while JC-1 staining demonstrated mitochondrial membrane depolarization, highlighting PEM-AuNPs’ ability to trigger mitochondrial dysfunction and apoptosis in NSCLC cells [240].

Lastly, biosynthesized gold nanoparticles (MLE-AuNPs) using Moringa oleifera leaf extract showed potent apoptotic effects in Dalton’s lymphoma cells [241]. MLE-AuNPs at 75 µg/mL induced G2/M phase cell cycle arrest by downregulating cyclin B1 and Cdc2 while upregulating p21 [241]. Flow cytometry analysis confirmed apoptosis induction, with Annexin V staining indicating apoptotic cell death. Gene expression analysis revealed increased levels of Bax, cytochrome c, and caspase-3, further confirming apoptotic pathway activation [241,242]. Importantly, MLE-AuNPs displayed selective cytotoxicity toward cancer cells without harming normal murine thymocytes, showcasing their potential as a sustainable and effective therapeutic option [241,243].

### 4.2. Gold Nanoparticles and Autophagy

Gold nanoparticles interact with cellular pathways in complex ways, and their role in autophagy modulation remains an area of ongoing investigation [244]. While research suggests that AuNPs can influence autophagic flux in cancer cells, the mechanisms remain incompletely understood due to the complexity of both nanoparticle behavior in biological environments and the dynamic nature of autophagy itself [245].

Xiaowei Ma et al. investigated the impact of AuNPs on autophagy, revealing their ability to disrupt normal autophagic processes [246]. Their study demonstrated that AuNPs are internalized into cells via endocytosis in a size-dependent manner and subsequently accumulate in lysosomes [246,247]. This accumulation leads to an increase in lysosomal pH, impairing normal lysosomal degradation [246,248]. Although AuNPs were found to induce autophagosome accumulation and LC3 processing, they also inhibited the degradation of the autophagy substrate p62, indicating a blockade of autophagic flux rather than true autophagy induction [246]. These findings suggest that AuNPs interfere with lysosomal function, with significant implications for their use in biomedical applications [246,249,250].

To further elucidate the relationship between AuNPs and autophagy, Li et al. studied the cellular response of MRC-5 human lung fibroblasts exposed to AuNPs. Their experiments showed that AuNPs triggered the formation of autophagosomes, confirmed by a significant upregulation of key autophagy-related proteins, including microtubule-associated protein 1 light chain 3 (MAP-LC3) and autophagy gene 7 (ATG7). Notably, the study also reported increased lipid peroxidation and oxidative stress, as evidenced by elevated malondialdehyde (MDA) protein adducts. The oxidative stress induced by AuNPs prompted cellular adaptation through the upregulation of antioxidant defense mechanisms. These findings highlight the bifunctional behavior of autophagy in nanoparticle interactions, functioning both as a protective response to oxidative stress and as a targetable pathway for therapeutic intervention [251,252].

Ma et al. further investigated how AuNPs contribute to autophagosome accumulation in treated cells [246]. Their findings emphasized that AuNPs disrupt lysosomal degradation capacity, primarily through alkalinization of lysosomal pH [246]. This impairment blocks the degradation of key autophagy substrates, such as p62, resulting in autophagosome accumulation [246]. The processing of LC3, a critical autophagosome marker, further confirmed the blockade of autophagic flux rather than enhanced autophagy induction [236,246].

Further experiments demonstrated that positively charged 50 nm AuNPs exacerbated the impairment of lysosomal function compared to negatively charged counterparts. The study reported a significant increase in autophagosome accumulation and lysosome enlargement in cells treated with positively charged AuNPs, as is presented in Figure 5. Confocal microscopy and transmission electron microscopy (TEM) provided visual confirmation of these morphological changes, reinforcing the hypothesis that AuNP charge plays a critical role in modulating autophagic pathways.

These findings underscore the complexity of AuNP interactions with autophagy, suggesting their potential to serve as modulators of autophagic processes in cancer therapy [244]. By disrupting autophagic flux and lysosomal function, AuNPs may sensitize cancer cells to treatment while simultaneously influencing cellular stress responses [244,253]. The implications of these interactions warrant further exploration, particularly regarding their impact on therapeutic efficacy and resistance mechanisms in cancer treatment.

#### Gold Nanoparticles as Radiosensitizers in Autophagy Induction

Recent studies indicate that gold nanoparticles possess promising characteristics as autophagy modulators, enhancing their therapeutic value in radiotherapy [8,146,254]. Alongside other nanomaterials, including silver nanoparticles, quantum dots, dendrimers, and rare earth oxide nanocrystals, AuNPs are now considered novel autophagy activators [255,256,257]. Their autophagy-inducing capacity is primarily attributed to increased ROS generation and subsequent mitochondrial impairment, which represent key initiators of autophagic signaling pathways [245,258,259]. However, the precise role and implications of autophagy modulation by AuNPs within the context of radiation therapy have yet to be fully elucidated [248]. Autophagy’s paradoxical roles—as both a protective mechanism and a cell-death mediator—render it a challenging therapeutic target [260].

Ma et al. explored the radiosensitizing efficacy of polyethylene glycol) (PEG)-coated gold nanospikes (GNSs) with various ligands, including amine (NH_2_), folic acid (FA), and cell-penetrating peptide (TAT) on KB cancer cells that overexpress folate receptors [261]. Their findings indicated a distinct correlation between surface functionalization and radiosensitization efficiency, following the order GNSs < NH_2_-GNSs < FA-GNSs < TAT-GNSs. The TAT-GNSs variant exhibited the highest sensitization enhancement ratio (SER = 2.30), linked to augmented ROS production, mitochondrial depolarization, and modified cell cycle distribution, as illustrated in Figure 6 [261].

Western blot analyses revealed increased levels of the autophagy marker LC3-II and the apoptosis marker activated caspase-3, alongside the accumulation of p62 proteins, signifying impaired autophagic flux. Importantly, the application of autophagy inhibitors further increased apoptosis levels, confirming a cytoprotective role of autophagy during radiation therapy. In vivo tests confirmed these results, demonstrating pronounced tumor growth suppression in response to combined X-ray and TAT-GNSs therapy, underscoring the therapeutic potential of dual autophagy–apoptosis modulation in cancer radiotherapy, as presented in Figure 7 [261].

In the same study, with the aid of flow cytometry, they further examined the complicated interplay between autophagy and apoptosis in AuNP-enhanced radiotherapy. Investigations on LC3-II expression revealed significantly elevated autophagic activity in cancer cells following treatment with GNSs and X-ray radiation. Interestingly, autophagy blockade using the inhibitor 3-methyladenine (3-MA) partially reversed this LC3-II increase, concurrently augmenting apoptosis, as evidenced by heightened levels of active caspase-3 and increased apoptosis rate from 11.05% (X-ray + GNSs) to 24.21% (X-ray + GNSs + 3-MA). These findings suggested that autophagy triggered by GNSs or X-ray exposure likely served as a protective response, mitigating apoptosis and thereby limiting radiotherapy efficacy. Consequently, combining autophagy inhibitors such as 3-MA with GNS-mediated radiotherapy may represent a viable strategy to enhance cancer cell radiosensitivity, shifting the balance toward apoptotic cell death [261].

### 4.3. Gold Nanoparticles as Drug Carriers in Autophagy Induction

#### 4.3.1. Autophagy Regulation by AuNPs-Based Chemotherapeutic Agents

As has been previously mentioned, autophagy in cancer cells is frequently activated as an adaptive survival mechanism under metabolic stress, though excessive autophagy can lead to cell death through lethal autophagy pathways. Recently, gold-based therapeutic agents have demonstrated promising outcomes by influencing these autophagic processes [262]. Maia et al. developed Au(III) thiosemicarbazone complexes that induced significant cytotoxicity in glioma cells by concurrently stimulating apoptotic and autophagic cell death [263]. Specifically, their study showed that a dual induction of apoptosis and lethal autophagy offers a compelling therapeutic strategy to combat glioma and potentially other aggressive malignancies [262,263].

Further advancing these findings, Zhang et al. synthesized novel Au(III) thiosemicarbazone compounds, notably compound C6, delivered via apoferritin nanoparticles (AFt-C6 NPs) engineered to effectively cross the blood–brain barrier (BBB) [262]. In vitro experiments using U87-MG glioma cells revealed that AFt-C6 nanoparticles markedly enhanced apoptosis and autophagy compared to free compounds [262]. Mechanistically, treatment with AFt-C6 NPs resulted in elevated LC3-II expression, reduced p62 levels, increased mitochondrial membrane depolarization, and elevated intracellular ROS, highlighting their ability to simultaneously activate lethal autophagy and apoptosis pathways [262]. These findings were corroborated by TUNEL staining and immunofluorescence analyses in vivo, confirming enhanced therapeutic efficacy in tumor models (Figure 8 and Figure 9) [262].

In a related study, Pandey et al. investigated the effect of AuNP size on therapeutic efficacy. Their study compared two different sizes of AuNPs—3.9 nm and 37.4 nm—in Lewis lung carcinoma (LLC) cells, demonstrating that larger AuNPs achieved greater cellular uptake, enhanced ROS generation, and improved radiosensitization [264]. Clonogenic assays confirmed that the larger AuNPs promoted higher rates of DNA damage, ultimately enhancing cell death through both apoptosis and autophagy pathways. These results emphasize the importance of nanoparticle size in modulating therapeutic responses and optimizing cancer treatment [264].

#### 4.3.2. Autophagy Enhancement by AuNPs-Based Drug Delivery

AuNPs offer significant advantages as drug carriers in chemotherapy by simultaneously modulating autophagic pathways [9,86]. Nanoparticle-mediated drug delivery can enhance drug solubility, stability, targeted uptake, and bioavailability. Notably, Pandey et al. demonstrated that size-dependent uptake of AuNPs directly influences therapeutic outcomes [264]. Utilizing Lewis lung carcinoma (LLC) cells, they found that larger AuNPs (approximately 37 nm) exhibited enhanced cellular uptake, significantly increased DNA damage, and higher sensitization effects compared to smaller particles [86,264]. These AuNPs amplified ROS-induced stress responses, ultimately modulating autophagy and apoptosis to achieve enhanced cytotoxicity in cancer cells, demonstrating a clear size-dependent relationship between nanoparticle properties and therapeutic efficacy [86,222,265].

In parallel, the work of Gharoonpour A et al. revealed that gold nanoparticles conjugated with natural compounds, such as dihydroartemisinin (DHA), significantly promoted cytotoxic effects against breast cancer cells by activating autophagy and apoptosis [266]. They observed increased LC3-II conversion and higher ROS generation in treated cells, which contributed to enhanced therapeutic outcomes compared to unconjugated agents. Importantly, the selectivity of these AuNP–drug complexes significantly reduced toxicity towards normal cells, further supporting their potential in cancer treatment strategies [178,266].

Expanding on nanoparticle functionality, researchers have explored the use of AuNPs to improve the delivery of genetic tools in cancer therapy [255,256]. While some systems involve AuNP-mediated delivery of small interfering RNA (siRNA), short-hairpin RNA (shRNA), or CRISPR/Cas9 targeting autophagy-related genes such as ATG12, ATG5, or ATG16L1, these approaches are primarily used to inhibit autophagy, particularly in cases where excessive autophagy contributes to tumor survival. As such, they are not discussed in detail here, as this section focuses on strategies aimed at enhancing autophagy. Nonetheless, the ability of AuNPs to facilitate gene delivery remains a promising therapeutic platform, especially in overcoming intracellular delivery barriers and improving treatment precision [77,257,258,259].

Recent research emphasizes that AuNPs influence epigenetic and molecular mechanisms critical for autophagy modulation and apoptosis in cancer treatment [260,267]. AuNPs can alter DNA methylation, histone acetylation, and miRNA expression patterns, significantly affecting autophagy-related genes such as Beclin-1, ATG5, and LC3 [261,262]. Several studies have reported that AuNPs upregulate these autophagy markers and enhance autophagic activity. For instance, Zhang et al. demonstrated that PEG-AuNPs increased LC3 and Beclin-1 levels while modulating autophagic flux in tumor-associated macrophages [268]. Similarly, Zhang et al. described nanoparticle-induced upregulation of LC3-II and Beclin-1 as central to autophagy induction in breast cancer cells [268]. These findings suggest that properly designed AuNPs may enhance autophagy through transcriptional or post-translational regulation of key proteins. However, the effects of AuNPs on autophagy are context-dependent. Certain formulations can impair lysosomal function, leading to autophagosome accumulation and disrupted autophagic flux, as shown by Ma et al. and López-Méndez et al. [246,267]. Such outcomes emphasize the importance of nanoparticle size, coating, and cellular context in determining biological effects. Overall, AuNP-mediated modulation of autophagy—when properly engineered—offers a promising strategy to enhance chemotherapeutic efficacy and target drug-resistant cancer cells [171].

To address potential toxicity associated with conventional AuNPs, researchers have designed biocompatible variants synthesized using natural compounds [269,270]. In one study, gold nanoparticles coated with deinoxanthin (DX), a tetraterpenoid from Deinococcus radiodurans, demonstrated enhanced anticancer efficacy in breast cancer models [271]. DX-coated AuNPs promoted ROS generation, triggering both apoptosis and autophagy pathways [86]. Importantly, these DX-AuNPs displayed superior biocompatibility, selectively targeting breast cancer cells with minimal toxicity to normal tissues [272]. This highlights the potential of natural compound-modified AuNPs as improved chemotherapeutic platforms [9].

Further research emphasizes the potential of AuNPs in co-delivery strategies, where nanoparticles carry both chemotherapeutic agents and autophagy modulators [184]. This dual-delivery approach enhances tumor cell sensitivity to chemotherapy by promoting apoptosis, DNA damage, and ROS-mediated autophagy [152]. Such strategies have been particularly effective in enhancing the efficacy of agents like cisplatin and gefitinib, offering an innovative direction for combination therapies [273].

### 4.4. Gold Nanoparticles in Dual Modulation of Apoptosis and Autophagy

AuNPs have, among others, shown potential as agents capable of inducing both apoptosis and autophagy, presenting a multifaceted approach to cancer therapy [155]. While these dual effects are still under investigation, emerging evidence suggests that AuNPs may simultaneously trigger these distinct cell death pathways, contributing to their therapeutic potential [155]. Understanding the mechanisms through which AuNPs modulate these processes remains an area of active research, yet recent studies provide compelling insights into their possible role in enhancing cancer treatment efficacy [38].

In an innovative study, researchers synthesized two organometallic gold(III) complexes, designated Cyc-Au-1 (AuL1Cl2, L1 = 3,4-dimethoxyphenethylamine) and Cyc-Au-2 (AuL2Cl2, L2 = methylenedioxyphenethylamine) [274]. These compounds feature C^N ligands resembling tetrahydroisoquinoline (THIQ) structures, which contribute to their potent anticancer activity [274]. Compared to cisplatin, both Cyc-Au-1 and Cyc-Au-2 demonstrated enhanced anticancer efficacy, lower resistance factors, and reduced toxicity in vitro [274].

Mechanistic investigations revealed that Cyc-Au-1 and Cyc-Au-2 predominantly accumulate in mitochondria, where they promote elevated reactive oxygen species production and endoplasmic reticulum (ER) stress. This mitochondrial dysfunction is known to trigger both apoptosis and autophagic cell death [275]. The concurrent activation of these pathways may enhance therapeutic outcomes by exploiting cellular stress responses [157]. Among the two compounds, Cyc-Au-2 showed superior therapeutic efficacy with lower toxicity in a murine tumor model [276]. Notably, Cyc-Au-2 is recognized as the first organometallic gold(III) compound reported to independently induce both apoptotic and autophagic cell death, highlighting its potential as a promising chemotherapeutic candidate [277,278].

In a separate investigation, researchers evaluated the role of AFt-C6 nanoparticles (AFt-C6 NPs) in modulating apoptosis and autophagy in glioma cells [262]. U87MG glioma cells were treated with AFt-C6 NPs in the presence or absence of the autophagy inhibitor 3-methyladenine (3-MA) to assess potential interactions between the two pathways [261,274]. After 48 h of treatment, caspase-3 expression levels remained comparable in both groups, regardless of 3-MA preincubation. This result suggests that the apoptosis and autophagy pathways activated by AFt-C6 NPs function independently without significant cross-regulation. The ability of AFt-C6 NPs to simultaneously trigger these independent mechanisms highlights their dual-mode action, contributing to enhanced therapeutic efficacy against glioma cells, as illustrated in Figure 10 [261,275,276,279].

In another study, Piktel et al. investigated the cytotoxic potential of gold nanopeanuts (AuP NPs) in ovarian cancer cells (SKOV-3) [222]. Using a range of experimental techniques, including colorimetric and fluorimetric assays, Western blotting, flow cytometry, and fluorescence microscopy, the researchers demonstrated that AuP NPs reduced cell viability and proliferation at concentrations ranging from 1 to 5 ng/mL over a 72 h treatment period [222]. The study identified ROS overproduction and, thus, both mitochondrial stress and cellular damage, which resulted in the concurrent activation of apoptosis and autophagy [222]. Western blot analysis confirmed the upregulation of pro-apoptotic proteins alongside increased expression of autophagic markers such as LC3-II [280,281]. These findings suggest that AuP NPs may exert potent cytotoxic effects through the combined modulation of apoptosis and autophagy, presenting a potential therapeutic option for ovarian cancer [222].

## 5. Discussion

Background research has made it increasingly evident that gold nanoparticles hold substantial promise in cancer therapy, owing to their unique physicochemical characteristics and ability to modulate critical cellular pathways [38,282]. Among these, apoptosis and autophagy have emerged as central mechanisms in determining cancer cell fate [91,283]. While apoptosis is a well-established form of programmed cell death triggered by genomic or metabolic stress, autophagy plays a more nuanced role—supporting cell survival under stress but also contributing to cell death under certain conditions [46,284].

This review examined the existing literature on how AuNPs influence these two essential pathways, both independently and through their interplay. Apoptosis and autophagy are not isolated processes; rather, their crosstalk contributes to tumor progression, therapeutic resistance, and treatment outcomes [34,285]. Understanding this dynamic relationship, particularly under the modulatory influence of AuNPs, offers a powerful approach to molecular-level intervention in cancer.

To structure this analysis, we explored three major domains where AuNPs have shown therapeutic relevance: as radiosensitizers enhancing the efficacy of radiation therapy; as drug carriers improving the precision and potency of chemotherapeutic delivery; and as dual modulators capable of concurrently activating or regulating both apoptosis and autophagy [156]. By evaluating these roles in the context of recent findings, we aimed to highlight the potential of AuNPs to overcome therapeutic resistance and improve treatment specificity while also identifying key areas requiring further research and clinical translation.

### 5.1. Current Research Progress and Observations

#### 5.1.1. Gold Nanoparticles in Radiation Therapy: Apoptosis and Autophagy Through Radiosensitization

In radiation oncology, the ability of gold nanoparticles to enhance tumor sensitivity to ionizing radiation represents one of the most compelling applications of nanomedicine in cancer therapy. Background research has indicated that mechanistically, AuNPs preferentially accumulate in critical organelles such as mitochondria and lysosomes, where they catalyze the generation of ROS upon exposure to IR [286]. Thereinafter, the elevated ROS levels promote mitochondrial membrane permeabilization, cytochrome c release, and downstream caspase activation, thereby initiating intrinsic apoptotic signaling [287]. Concurrently, ROS-induced lysosomal dysfunction and endoplasmic reticulum stress can trigger autophagy—a process that may either sensitize cancer cells to radiation-induced death or support survival depending on the cellular context [288].

Preclinical studies across multiple tumor types—including melanoma, glioma, colorectal, and lung cancers—have consistently demonstrated the radiosensitizing effects of AuNPs through intensified oxidative and genotoxic stress [197,289]. Furthermore, these studies also indicate that AuNP size, shape, surface charge, and functionalization critically modulate therapeutic efficacy [9,144]. Most importantly, while for many years now, enhanced DNA damage and apoptosis remain the most prominent outcomes, a subset of investigations has reported autophagy activation following IR-AuNP exposure, suggesting that both pathways are responsive to AuNP-mediated stress [290,291]. Although these findings remain primarily confined to in vitro and in vivo models, they collectively affirm the promise of AuNPs as effective radiosensitizers, particularly in radioresistant or hypoxic tumor microenvironments where standard therapies often fail [8,151].

#### 5.1.2. Gold Nanoparticles in Chemotherapy: Apoptosis and Autophagy Through Drug Delivery

In chemotherapeutic contexts, AuNPs serve as multifunctional agents, acting both as targeted drug delivery vehicles and active participants in apoptosis and autophagy modulation [9,38]. The literature indicates that their high surface area, tunable dimensions, and modifiable surfaces enable conjugation with chemotherapeutic agents—such as doxorubicin, cisplatin, pemetrexed, and paclitaxel—as well as with biologics like siRNA or peptides [38,292]. These conjugates benefit from improved stability, selective tumor accumulation, and controlled release in response to tumor-specific stimuli such as pH or redox changes [293].

Notably, various research teams observed that AuNPs themselves exhibit pro-apoptotic effects by influencing mitochondrial integrity and increasing ROS generation [205,294]. In parallel, studies have shown that AuNPs can disrupt autophagic flux or promote lethal autophagy depending on the cellular context and nanoparticle properties. Positively charged AuNPs, for instance, are more likely to induce autophagosome accumulation and lysosomal impairment, while particle size influences both uptake efficiency and intracellular trafficking [246]. Moreover, the epigenetic influence of AuNPs—altering histone modification and microRNA profiles—has been linked to shifts in apoptotic and autophagic gene expression, expanding their impact beyond drug transport alone [295].

Through this dual action, AuNPs enhance the efficacy of chemotherapeutic agents while simultaneously manipulating key survival and death pathways. This functionality positions them as next-generation therapeutic platforms capable of overcoming resistance mechanisms that limit the success of conventional chemotherapy [38,156].

#### 5.1.3. Dual Modulation of Apoptosis and Autophagy with Gold Nanoparticles: A Synergistic Approach

Perhaps the most promising and innovative frontier in AuNP research lies in their capacity to modulate both apoptosis and autophagy simultaneously [296]. The research shows that these two pathways—often viewed as antagonistic—are deeply intertwined, and their crosstalk significantly impacts cancer cell fate and treatment response. More precisely, AuNPs, especially those structured as gold(III) complexes or engineered with targeted delivery systems, have shown the ability to activate both mechanisms through shared upstream triggers [238,297].

Furthermore, compounds like Cyc-Au-2 and AFt-C6 nanoparticles have demonstrated concurrent activation of caspase-mediated apoptosis and LC3-II-mediated autophagy in glioma and other aggressive cancers [298,299]. Importantly, these responses appear to occur independently, suggesting a lack of inhibitory crosstalk and allowing both pathways to contribute to therapeutic cytotoxicity [300,301]. In cases where apoptotic machinery is defective, autophagy induction may provide an alternative route to tumor cell death. Conversely, when autophagy acts as a protective mechanism, co-administration of autophagy inhibitors has been shown to enhance AuNP-induced apoptosis, emphasizing the therapeutic potential of pathway co-modulation [283].

This dual-action strategy may be particularly valuable in treating tumors that exhibit resistance to single-mode therapies [302,303]. By engaging two distinct cell death modalities, AuNPs provide a robust tool for overcoming the heterogeneity and plasticity of cancer cells, underscoring their relevance in the design of future combination regimens [38,304].

Table 1 summarizes key findings from recent studies investigating the therapeutic and biological effects of various gold nanoparticle formulations. It details the specific nanoparticle types used, highlighting their roles in inducing apoptosis and autophagy, along with other observed biological effects and molecular mechanisms involved.

To better understand current research trends and identify underexplored areas in the field, we constructed an indicative mapping of the literature across apoptosis, autophagy, and AuNP-based cancer therapies, with a focus on chemotherapy and radiotherapy (Figure 11). This visual overview reflects our comprehensive review of published studies. It should be noted that this is not a systematic review; the literature search is non-quantitative and intended to provide an indicative representation of trends. Apoptosis remains the most extensively studied mechanism across therapeutic modalities, particularly in combination with AuNPs. In contrast, autophagy—while increasingly recognized for its multifaceted role in tumor suppression and resistance—remains comparatively less investigated, especially in the context of nanotechnology-enhanced therapies.

A notable imbalance is evident between in vitro and in vivo models, with limited translational studies bridging preclinical findings to complex biological systems. Moreover, clinical investigations involving AuNPs remain extremely scarce, underscoring a critical bottleneck in translational oncology. The co-modulation of apoptosis and autophagy by AuNPs in therapeutic contexts—though mechanistically intriguing—has yet to be explored in depth across most cancer types.

These observations point toward meaningful gaps and limitations in current literature that future studies could address, particularly regarding autophagy’s functional duality in AuNP-mediated therapy and the need for robust in vivo and clinical validation. As such, this tripartite interaction between AuNPs, apoptosis, and autophagy warrants continued interdisciplinary focus to realize its full therapeutic potential.

### 5.2. Limitations

Despite the growing body of research supporting the anticancer potential of AuNPs, several limitations persist. One of the most pressing concerns is the paucity of clinical trials. While in vitro and in vivo studies provide compelling evidence that AuNPs can modulate apoptosis and autophagy to improve cancer outcomes, these effects remain largely unverified in clinical contexts. The complexity of translating bench-side findings to bedside application is compounded by challenges related to biodistribution, long-term retention, immunogenicity, and ethical acceptability, all of which demand comprehensive evaluation before AuNPs and personalized molecular therapies can be considered clinically viable.

In general, the heterogeneity in nanoparticle design across studies reflects the inherently versatile nature of nanoscale systems. Variations in AuNP size, morphology, surface chemistry, and functional coatings are not simply methodological divergences but manifestations of a broader complexity arising from dynamic physicochemical interactions, such as Brownian motion, electromagnetic forces, and variable surface energy states. These interactions, coupled with the biological variability of the tumor microenvironment, render nanoparticle behavior highly context-dependent and often unpredictable. As such, even minor alterations in synthesis parameters can drastically shift cellular uptake, biodistribution, and therapeutic efficacy. This experimental plasticity, while offering opportunities for precision design, also poses significant challenges for reproducibility, safety assessment, and clinical translation.

Limitations are not confined to the nanoparticles themselves. Preclinical models—ranging from conventional 2D and advanced 3D cell cultures to in vivo animal systems—have been instrumental in the preliminary evaluation of gold nanoparticle-based therapies and the standardization of methodological approaches. These models provide controlled environments for assessing cytotoxicity, cellular uptake, and mechanistic pathways such as apoptosis and autophagy. However, their predictive capacity remains limited when it comes to replicating the intricate physiology of human disease. Most in vitro systems fail to reflect the architectural, stromal, and immunological complexity of solid tumors, while commonly used animal models—particularly immunodeficient mice—do not emulate human pharmacokinetics, immune competence, or tumor heterogeneity.

The biological pathways targeted by AuNPs—specifically apoptosis and autophagy—introduce additional complexity. These mechanisms are deeply interconnected and highly context-dependent. In some tumor types, autophagy acts as a protective process that enables cell survival under stress, while in others, it contributes to therapeutic cell death. Similarly, apoptosis regulation is frequently altered in treatment-resistant cancers, making its induction by AuNPs unpredictable. Without detailed molecular profiling of these pathways in specific cancers, therapeutic interventions risk unintended effects, including resistance or adaptation.

The intertumoral heterogeneity of cancers further complicates AuNP-based approaches. Variations in apoptotic sensitivity, autophagic flux, and signaling plasticity are observed not only between cancer types but also between patients and even within a single tumor. Cancers such as glioblastoma, pancreatic adenocarcinoma, and triple-negative breast cancer often exhibit high levels of resistance to conventional therapies, and strategies that enhance apoptosis alone may be insufficient. Additionally, biological barriers such as the blood–brain barrier (BBB) in central nervous system malignancies present substantial obstacles to the effective delivery of AuNPs and associated therapeutics.

Moreover, while AuNPs are frequently described as biocompatible, concerns regarding chronic toxicity and off-target accumulation remain unresolved. Studies have documented nanoparticle retention in organs such as the liver, spleen, and kidneys, raising questions about long-term safety and the potential for cumulative damage. Surface functionalization, while beneficial for targeting and circulation, may also provoke immune activation or hypersensitivity reactions, particularly with repeated dosing.

Finally, ethical constraints associated with animal experimentation, including welfare concerns and regulatory limitations, continue to challenge the design and implementation of in vivo studies. However, these considerations are further magnified during the transition to clinical translation as the bioethical landscape becomes significantly more complex when therapies involve nanotechnology that interacts with cellular and molecular machinery. The potential for off-target effects, unpredictable biodistribution, and long-term accumulation of AuNPs raises critical questions about patient safety, informed consent, and long-term monitoring. In this context, AuNP-based therapies—though promising—confront new dimensions of ethical inquiry, including the manipulation of cell death pathways, unintended interference with immune or regenerative systems, and the challenge of communicating risk in a field where long-term outcomes are still poorly understood.

#### Clinical Translation and Regulatory Considerations

While numerous preclinical studies investigating AuNPs in the context of apoptosis and autophagy have demonstrated promising results, clinical translation remains minimal. To date, the clinical implementation of AuNP-based therapies has been extremely limited [9]. A notable example includes a first-in-human clinical trial (NCT01270139) evaluating the safety and feasibility of gold nanoshell-mediated photothermal ablation, which reported a favorable safety profile and represented an initial proof-of-concept in humans [305]. Similarly, AuroShells—commercially developed gold nanoparticles—have been tested in a small-scale clinical study for prostate cancer treatment, demonstrating technical feasibility and localized control. However, significant translational challenges still persist. As detailed in the previous section, biodistribution and long-term organ retention remain key concerns for safety, particularly regarding accumulation in the liver, spleen, and kidneys [306,307]. While certain formulations, such as glutathione-coated ultrasmall AuNPs, have shown improved clearance and reduced toxicity, comprehensive pharmacokinetic profiling across diverse nanoparticle designs is still lacking [289].

Crucially, while many preclinical studies have demonstrated the capacity of AuNPs to modulate key cell death pathways—namely, apoptosis and autophagy—the clinical implications of these effects remain largely unexplored [39]. The absence of clinical trials directly evaluating these mechanistic pathways underscores a significant translational gap. Bridging this gap will require not only efficacy and safety validation but also mechanistic biomarkers that can link AuNP exposure to pathway-specific therapeutic outcomes in human tumors [308,309].

On the regulatory front, the lack of harmonized standards for evaluating the efficacy and safety of nanomedicine products complicates approval [310]. AuNPs exhibit unique interactions at the molecular and cellular levels, which traditional drug evaluation frameworks may not fully capture [9]. As such, the development of tailored regulatory guidelines is essential to advance clinical adoption.

Another central issue is patient selection. Tumor heterogeneity significantly influences AuNP uptake and the expression of apoptotic and autophagic machinery [311]. This makes stratification strategies critical for future trials. Recent studies underscore the value of predictive biomarkers and non-invasive imaging tools—such as radiomics and functional imaging—to estimate AuNP accumulation and identify tumors likely to respond based on their apoptotic or autophagic profiles [312,313]. These techniques hold the potential for enabling precision nanomedicine by identifying patients most likely to benefit from AuNP-based interventions targeting these pathways.

### 5.3. Future Directions

Future research on AuNPs in cancer therapy—particularly their role in modulating apoptosis and autophagy—holds transformative potential. Crucially, the mechanistic linkage between apoptosis and autophagy remains insufficiently understood, especially in the context of nanoparticle-based interventions. As both pathways dynamically interact to regulate cell fate, a deeper mechanistic understanding of how AuNPs influence this crosstalk is essential for advancing therapeutic precision.

A central avenue for advancement involves the identification of molecular biomarkers that can predict tumor responsiveness to AuNP-induced apoptotic and autophagic signaling. Studies have shown that tumor heterogeneity can significantly affect therapeutic outcomes, with some cancers favoring autophagy-mediated survival and others more susceptible to apoptosis [314,315]. Stratifying tumors based on autophagy and apoptosis pathway activity could enable patient-specific approaches, ensuring optimal responses to AuNP-based therapies.

Equally, combination strategies that integrate AuNPs with autophagy inhibitors—such as 3-methyladenine (3-MA) or chloroquine—represent a promising direction. Since autophagy can serve as a resistance mechanism in apoptosis-primed cancer cells, targeted inhibition may sensitize tumors to AuNP-induced apoptotic stress [316]. This is particularly relevant in hypoxic or nutrient-deprived tumors, where autophagy contributes to adaptive survival.

Moreover, the design of next-generation AuNPs capable of simultaneously activating both autophagic and apoptotic death pathways could provide a robust strategy against therapy-resistant cancers. Research into gold(III) complexes and peptide-functionalized AuNPs has shown that co-activation of these pathways may bypass resistance mechanisms that limit the efficacy of single-pathway interventions [38]. However, balancing the dual activation remains a critical challenge, as autophagy’s role is highly context-dependent.

To support clinical advancement, future studies must also prioritize comprehensive in vivo evaluations, addressing biodistribution, clearance, and long-term biocompatibility. Advanced preclinical models—such as patient-derived organoids and immune-competent xenografts—should be employed to better mimic human tumor microenvironments and immune responses. Additionally, systems biology and computational modeling can assist in optimizing nanoparticle design, forecasting pathway interactions, and tailoring therapeutic windows [317].

Finally, integrating AuNPs into multi-modal treatment regimens, such as radiotherapy, chemotherapy, and immunotherapy, represents a compelling clinical strategy. AuNPs have shown synergistic potential in enhancing DNA damage during ionizing radiation and improving drug accumulation in tumors. When coupled with the modulation of apoptotic and autophagic pathways, these combinatory approaches may significantly improve outcomes in aggressive or treatment-refractory cancers. However, to fully exploit AuNPs’ radiosensitizing capacity, further investigation is needed into their behavior under clinically relevant megavoltage (MV) energy levels. While current evidence predominantly supports their efficacy in the kilovoltage (kV) range—where the photoelectric effect is dominant and energy deposition is maximized—clinical radiotherapy typically operates at MeV energies, where Compton scattering prevails and dose enhancement by AuNPs may be attenuated. Bridging this energy gap remains essential to translating preclinical findings into effective clinical protocols.

## 6. Conclusions

In summary, this review underscores the critical and evolving role of AuNPs in modern oncology, particularly through their unique capacity to interface with the molecular dynamics of apoptosis and autophagy. Acting as active biological agents, AuNPs have demonstrated the ability to modulate cell death and survival pathways with remarkable specificity. Whether used to enhance the efficacy of ionizing radiation or to facilitate targeted drug delivery, their dual function positions them at the forefront of next-generation cancer therapeutics.

Central to this investigation was the question of whether apoptosis and autophagy—two fundamental yet intricately linked mechanisms—can be jointly modulated by AuNPs for therapeutic gain. The evidence presented suggests that these pathways do not operate in isolation but instead exist in a delicate interplay, one that AuNPs are uniquely poised to influence. By shifting the cellular balance between survival and death, AuNPs offer a molecularly informed strategy to overcome tumor resistance, particularly in aggressive or refractory malignancies.

Crucially, this tripartite framework—apoptosis, autophagy, and nanotechnology—introduces a new paradigm for cancer treatment. Rather than relying on dose escalation or systemic toxicity, this approach enables precise biological targeting and reprogramming of cancer cell fate. Moreover, it opens the door to personalized interventions through the integration of molecular biomarkers, tailored nanoparticle engineering, and multimodal therapy combinations.

While considerable challenges remain—particularly in translating preclinical promise into clinical efficacy—the convergence of these three domains marks a pivotal shift in therapeutic philosophy. It redefines the battlefield of cancer not merely as a site of destruction but as a space of strategic molecular negotiation. As research advances, harnessing the synergy of this intricate triptych holds the transformative potential to reshape the future of cancer therapy into one that is more effective, adaptable, and profoundly patient-centered.

## Figures and Tables

**Figure 1 cimb-47-00460-f001:**
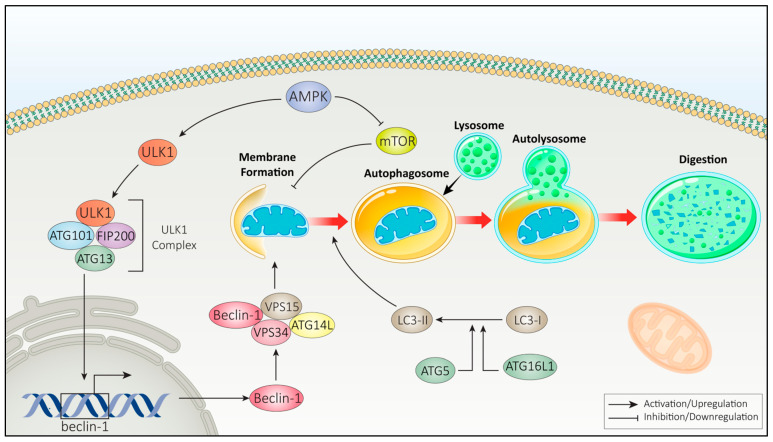
I lipidates to LC3-II, aided by ATG5/ATG16L1. LAMP2 and SNAREs mediate fusion, enabling cargo degradation in autophagosome–lysosome, Elsevier License Number: 5997550443775 [86].

**Figure 2 cimb-47-00460-f002:**
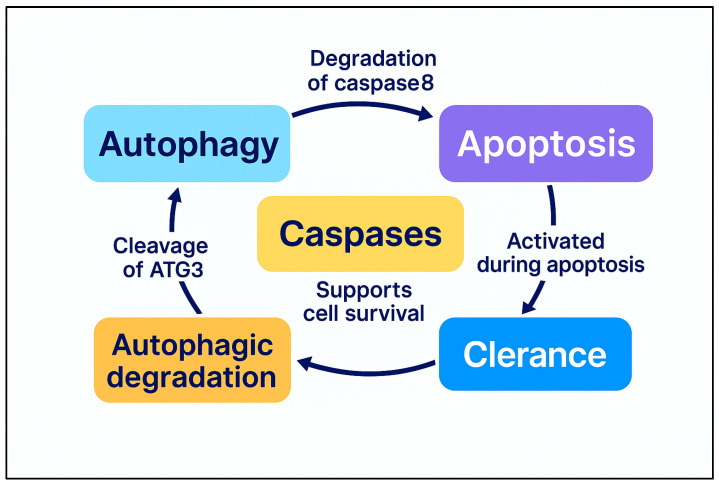
Schematic illustration of the crosstalk between autophagy and apoptosis. This diagram highlights the core regulatory mechanisms linking autophagy and apoptosis. Caspases, central to apoptotic execution, inhibit autophagy by cleaving key proteins such as ATG3. Autophagy, in turn, can suppress apoptosis by degrading pro-apoptotic factors like caspase-8 and maintaining cellular homeostasis through the clearance of damaged organelles and protein aggregates. Impaired or excessive autophagy may instead promote apoptosis.

**Figure 3 cimb-47-00460-f003:**
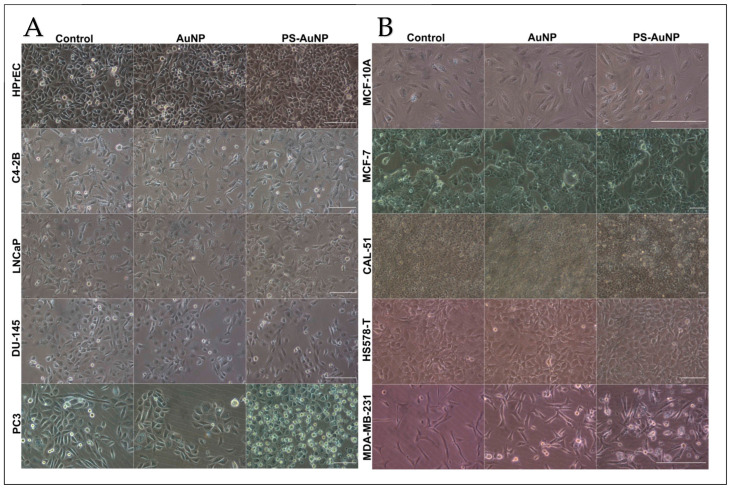
Morphological analysis of (**A**) prostate (HPrEC, C4-2B, LNCaP, DU-145, and PC3) and (**B**) breast (MCF-10A, MCF-7, CAL-51, HS578-T, and MDA-MB-231) cell lines following exposure to PBS (control), gold nanoparticles (AuNPs), or PS-functionalized AuNPs (PS-AuNPs). Representative light microscopy images show cell morphology changes across treatments. Notably, cells exposed to PS-AuNPs (right column) exhibit pronounced morphological alterations such as cell shrinkage and detachment, characteristic of apoptotic activity. Scale bar: 50 µm. Copyright: © 2021 by the authors. Licensee MDPI, Basel, Switzerland. This article is an open access article distributed under the terms and conditions of the Creative Commons Attribution (CC BY) license [235].

**Figure 4 cimb-47-00460-f004:**
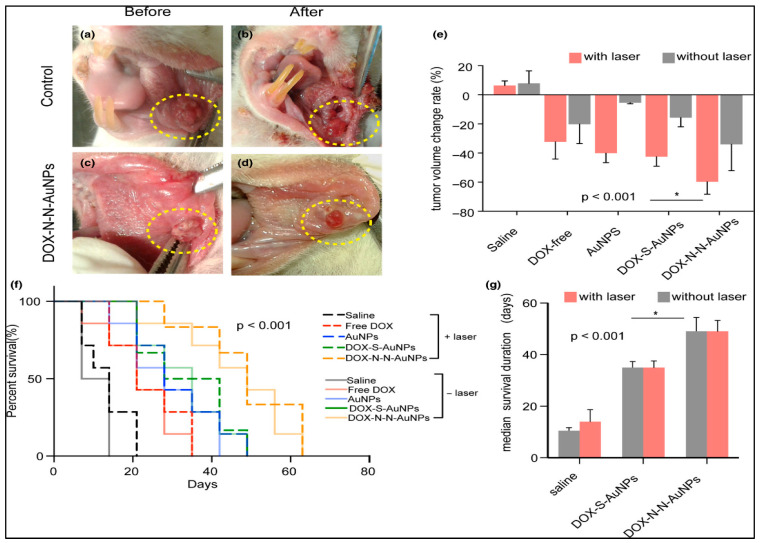
Clinical images illustrate tumor progression before (**a**,**b**) and after treatment (**b**,**d**) with DOX-AuNPs. After 4 weeks, the saline-treated group (**a**,**b**) showed little to no tumor reduction, whereas the group receiving DOX-N-N-AuNPs intralesionally combined with laser (**c**,**d**) exhibited a marked reduction in tumor size. (**e**) A bar graph presents the average percentage change in normalized tumor volume across IL-treated groups (with and without laser) 4 weeks post-treatment. Notably, all groups treated with AuNPs, particularly those also receiving laser therapy, demonstrated substantial tumor shrinkage compared to the saline control. The most significant tumor reduction was observed in the DOX-N-N-AuNPs plus laser group (*p* < 0.001). (**f**) Kaplan–Meier survival analysis of IL-treated animals indicates a significantly extended lifespan in hamsters treated with DOX-N-N-AuNPs, both with and without laser application (*p* < 0.001). (**g**) Median survival time increased to 50 ± 4.7 days in animals administered DOX-N-N-AuNPs, in contrast to 34 ± 3.6 days in those treated with DOX-S-AuNPs (*p* < 0.001). Data are presented as mean or median ± standard deviation. Statistical significance is denoted by * (*p* < 0.001). Copywrite: © 2020 John Wiley & Sons A/S. Published by John Wiley & Sons Ltd. All rights reserved.

**Figure 5 cimb-47-00460-f005:**
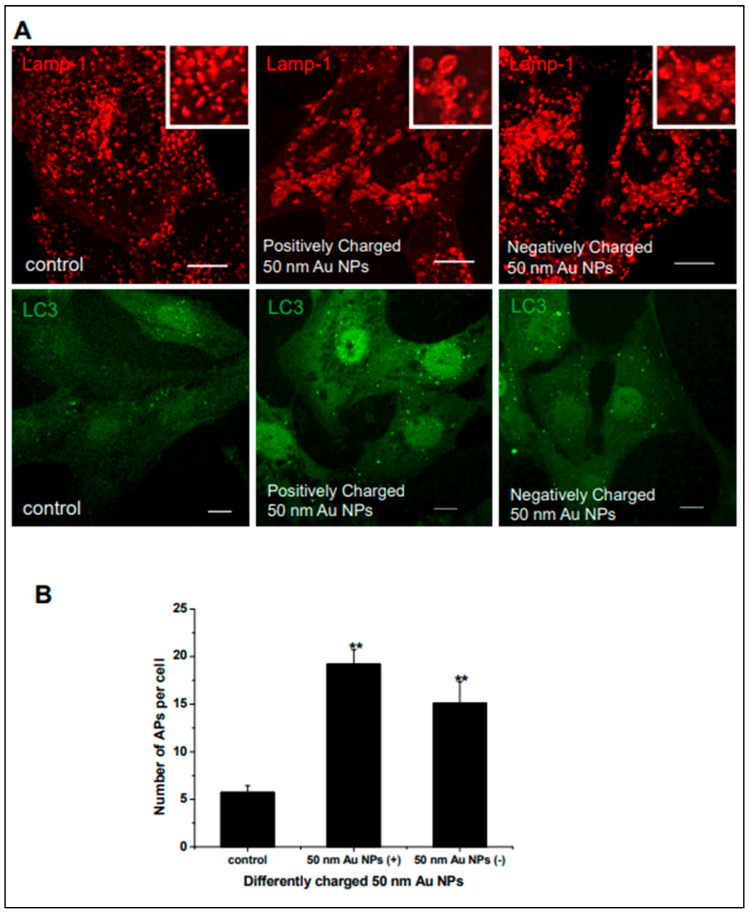
LC3 puncta formation and lysosomal enlargement following exposure to differently charged gold nanoparticles: (**A**) Confocal microscopy images of NRK cells stably expressing Lamp1-RFP (red) and LC3-CFP (shown in green) were captured following a 24 h treatment with either positively or negatively charged 50 nm AuNPs. Enlarged views are provided in the insets. (Scale bar: 10 µm.) (**B**) Quantitative analysis showing the average number of autophagosomes per cell after 24 h of nanoparticle exposure. Permission/license is granted, Copyright © 2011, American Chemical Society [246].

**Figure 6 cimb-47-00460-f006:**
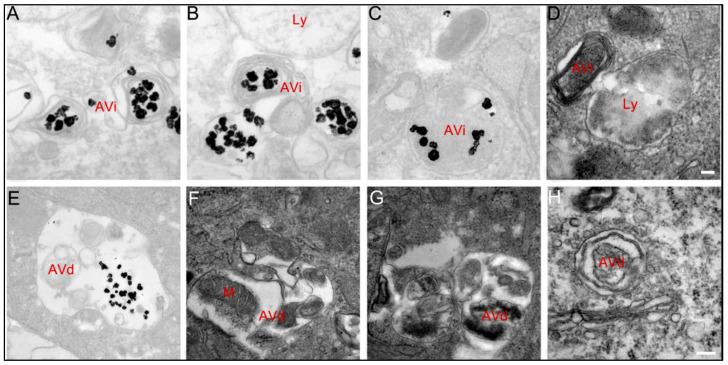
Biological transmission electron microscopy (TEM) images illustrating the ultrastructure of autophagosomes and autophagic vacuoles in KB cells treated with TAT-functionalized gold nanostars (TAT-GNSs; 50 μg/mL Au) combined with X-ray irradiation (4 Gy). (**A**–**D**) Early-stage autophagic vacuoles (AVi) and autophagosomes characterized by multilamellar or double-membrane structures. Scale bar: 100 nm. (**E**–**H**) Late-stage or degradative autophagic vacuoles (AVd), featuring partially degraded organelle contents, such as mitochondria (M), indicating advanced stages of autophagy. Ly: lysosome; M: mitochondria; AVd: late-stage autophagic vacuole. Scale bar: 200 nm. Images reproduced with permission. Copyright © 2017, American Chemical Society.

**Figure 7 cimb-47-00460-f007:**
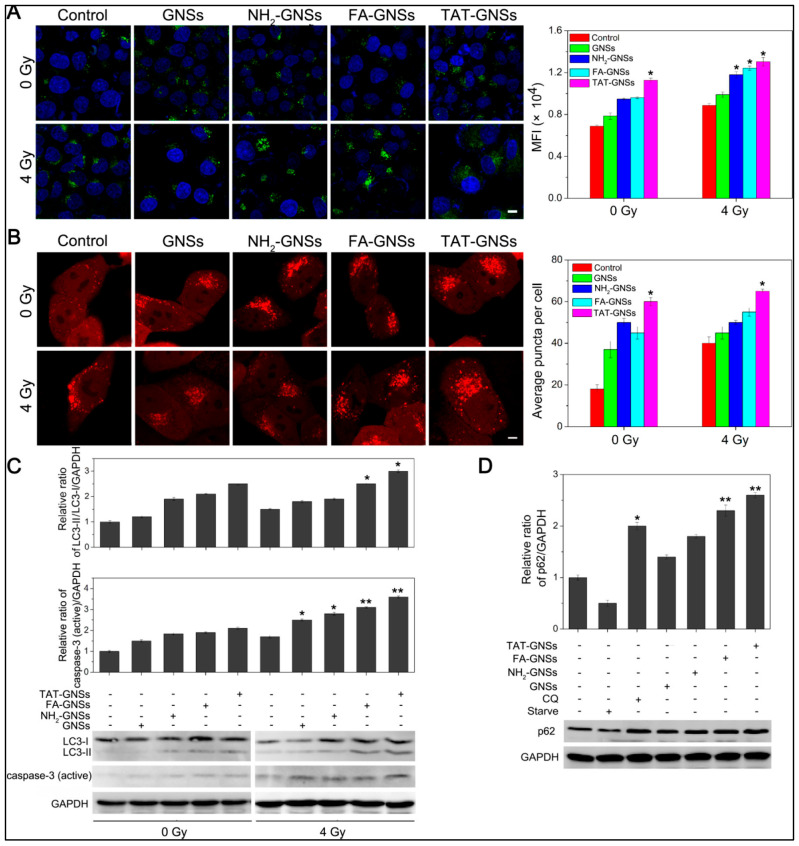
Autophagy induction in KB cells exposed to surface-modified gold nanostars (GNSs). (**A**) Confocal fluorescence microscopy images of KB cells untreated (control) or treated for 24 h with various surface-functionalized GNSs—unmodified GNSs, amine-functionalized (NH2-GNSs), folic acid-conjugated (FA-GNSs), or TAT-peptide-modified (TAT-GNSs)—at 50 μg/mL Au, followed by X-ray irradiation (4 Gy). Nuclei were stained blue (Hoechst 33342), and autophagosomes/autolysosomes were stained green (Cyto-ID dye). (**B**) Confocal images of KB cells expressing RFP-LC3, illustrating autophagy through LC3 puncta formation (red dots), with quantification (average puncta/cell). Scale bars: (**A**) 10 μm; (**B**) 5 μm. (**C**) Western blot analysis of LC3-I, LC3-II, and caspase-3 proteins post-treatment, with GAPDH as control. (**D**) Western blot showing p62 protein levels under various conditions, including starvation and chloroquine treatment. Symbols (*, **) indicate significant differences (*p* < 0.05 and *p* < 0.01, respectively) compared to untreated controls. Permission/license is granted, Copyright © 2017, American Chemical Society [261].

**Figure 8 cimb-47-00460-f008:**
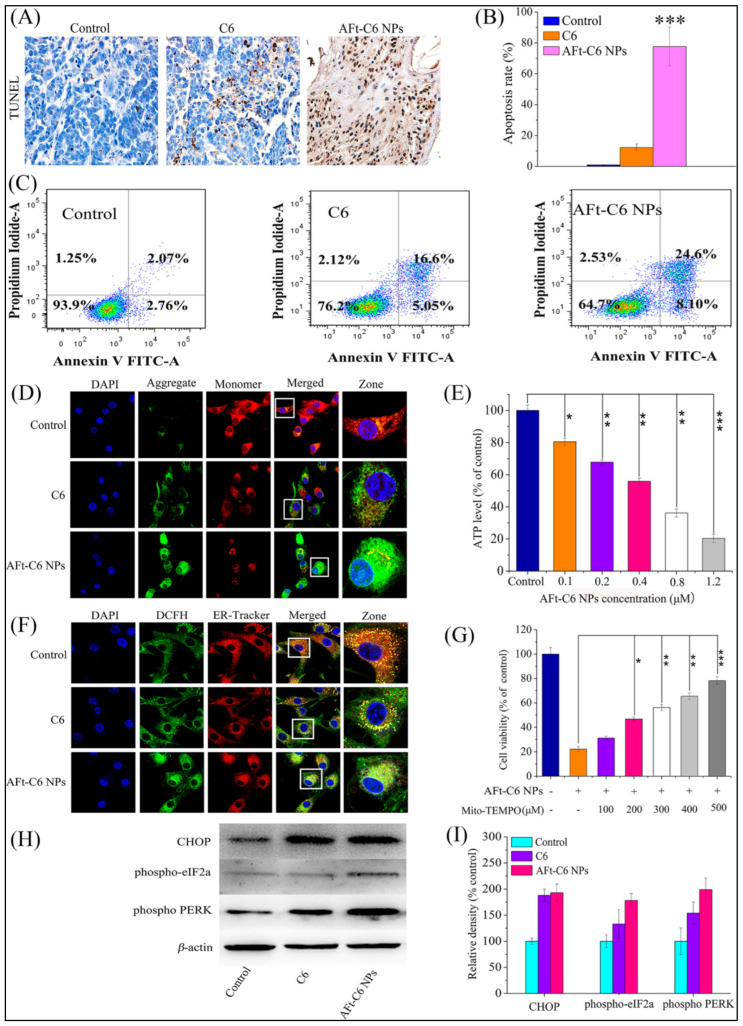
In vitro and in vivo analysis of apoptosis induced by C6 and AFt-C6 nanoparticles. (**A**) TUNEL staining of tumor tissues showing apoptotic cells (scale bar: 50 μm). (**B**) Quantification of TUNEL-positive cells. (**C**) Apoptosis in U87MG cells after 24 h treatment with C6 or AFt-C6 nanoparticles (0.3 μM); colors (blue, green, orange) indicate live, early apoptotic, and late apoptotic cells, respectively. (**D**) JC-1 staining demonstrating mitochondrial membrane potential changes; white rectangles indicate magnified areas highlighting specific mitochondrial regions (scale bar: 10 μm). (**E**) Intracellular ATP levels in U87MG cells after AFt-C6 NP treatment for 24 h. (**F**) ROS (green) and ER (red) levels in cells treated with C6 or AFt-C6 nanoparticles; white rectangles highlight areas enlarged to detail ROS and ER interactions (scale bar: 10 μm). (**G**) Effect of mitochondrial superoxide scavengers on AFt-C6-induced cell death. (**H**) Western blot analysis of ER stress markers. (**I**) Relative expression of ER stress proteins compared to controls. Symbols (*, **, ***) indicate significant differences (*p* < 0.05, *p* < 0.02, and *p* < 0.001, respectively). Permission/license is granted, Copyright © 2020, American Chemical Society [262].

**Figure 9 cimb-47-00460-f009:**
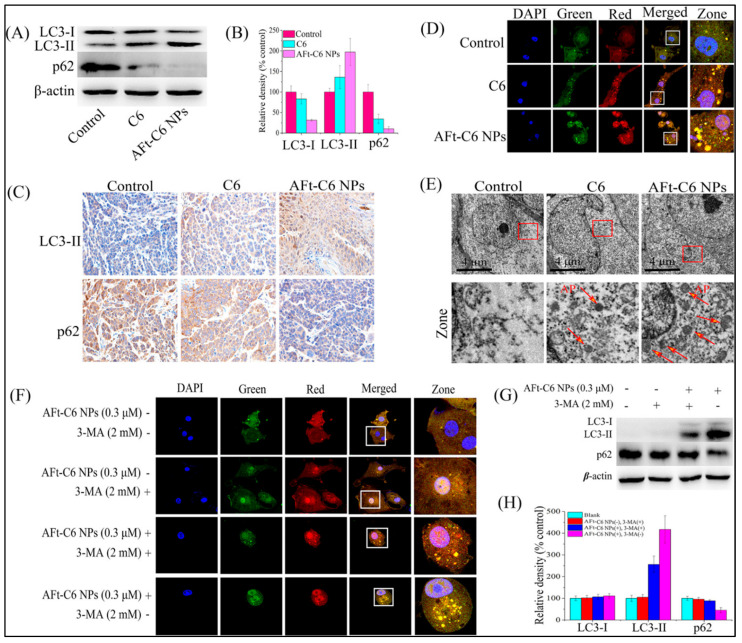
In vitro and in vivo evaluation of lethal autophagy induced by C6 and AFt-C6 nanoparticles (NPs) in U87MG cells. (**A**) Western blot analysis of autophagy-related proteins (LC3-II, p62, Beclin-1, and ATG5) following treatment with C6 or AFt-C6 NPs. (**B**) Quantitative analysis of relative expression levels of autophagy proteins normalized to control (%). (**C**) Immunohistochemical staining images of LC3-II and p62 in tumor tissue sections (scale bar = 50 µm). (**D**) Immunofluorescence images of U87MG cells transiently transfected with RFP-GFP-LC3-II plasmid and treated with C6 or AFt-C6 NPs for 24 h. Yellow puncta represent autophagosomes, red puncta indicate autolysosomes (scale bar = 20 µm). (**E**) Transmission electron microscopy (TEM) images displaying autophagic vesicle formation in U87MG cells post-treatment (scale bar = 500 nm). (**F**) Immunofluorescence imaging of U87MG cells pretreated with the autophagy inhibitor 3-MA, followed by exposure to AFt-C6 NPs, showing the inhibition of autophagosome formation (scale bar = 20 µm). (**G**) Western blot confirming suppression of autophagy markers by 3-MA treatment. (**H**) Quantitative analysis of autophagy marker expression levels relative to control. Note: Rectangles in subfigures (**D**–**F**) highlight regions of interest illustrating characteristic autophagic structures. All images have been updated to meet minimum resolution standards (≥1000 pixels width/height, ≥300 dpi). Permission/license is granted, Copyright © 2020, American Chemical Society [262].

**Figure 10 cimb-47-00460-f010:**
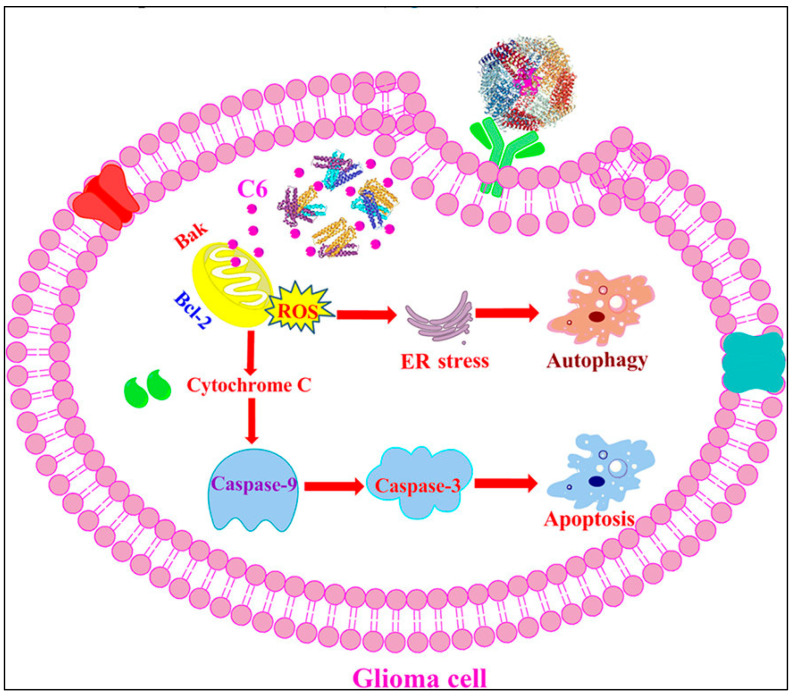
Designing a novel gold(III)-based therapeutic for glioma using apoferritin nanoparticles: dual induction of lethal autophagy and apoptosis. Permission/license is granted, Copyright © 2020, American Chemical Society [262].

**Figure 11 cimb-47-00460-f011:**
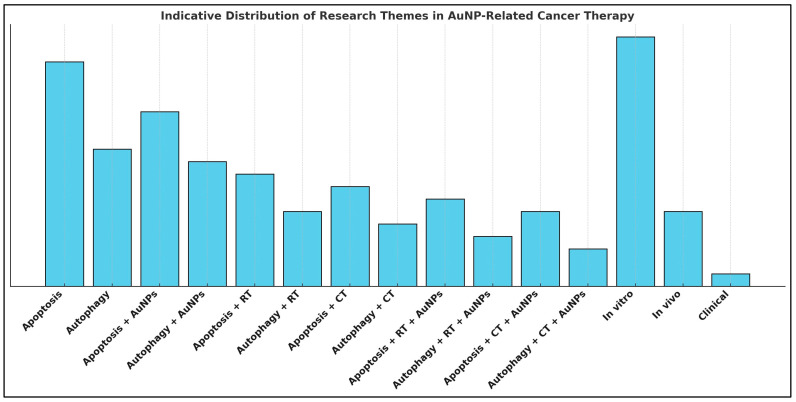
Distribution of the literature across key research domains involving gold nanoparticles (AuNPs) in cancer-related apoptosis and autophagy. This bar plot provides an indicative overview of studies published between 1970 and 2025 across selected thematic categories, including apoptosis, autophagy, radiosensitization, drug delivery, chemotherapy (CT), and radiotherapy (RT), as well as their intersections with AuNP-based strategies. The plot also includes the distribution of in vitro, in vivo, and clinical investigations.

**Table 1 cimb-47-00460-t001:** Summary of key findings from recent basic research investigating apoptosis and autophagy mechanisms induced by AuNPs for applications in cancer treatment.

Study	Nanoparticle Type	Key Findings	Mechanisms Observed
Bemidi-nezhad et al. [196]	Glucose-coated AuNPs (Glu-GNPs), liposome-encapsulated gold ions (Gold-Lips)	Enhanced ROS production, mitochondrial dysfunction, apoptosis; improved radiosensitivity in melanoma cells	Enhanced Bax, p53, Caspase-3/-7; downregulated Bcl-2
Tsai et al. [198]	55 nm AuNPs	Enhanced ROS accumulation, mitochondrial dysfunction, apoptosis in epidermoid carcinoma cells post γ-ray exposure	Increased ROS levels, mitochondrial damage, caspase activation
Saberi et al. [203]	AuNPs with 9 MV radiation	Increased apoptosis in HT-29 colorectal cancer cells	Increased DNA damage, mitochondrial dysfunction
Neshast-ehriz et al. [206]	Gold-coated iron oxide nanoparticles (Au@IONPs)	Enhanced apoptosis via ROS-mediated damage, mitochondrial stress; synergistic effects with hyperthermia and X-ray	Significant apoptotic markers increase, reduced cell viability
Hu et al. [210]	AuNP-pep@Mem	Developed probe detecting apoptosis via caspase-3; real-time tracking of apoptotic events	Caspase-3 activation, mitochondrial membrane disruption
Surapaneni et al. [220]	Citrate-capped and cysteamine-capped AuNPs	Citrate-AuNPs increased histone deacetylation, induced apoptosis; cysteamine-AuNPs activated p38 MAPK pathway	Differential gene regulation (caspase-3, Bax, Bcl-2)
Piktel et al. [222]	Gold nanopeanuts (AuP NPs)	Induced apoptosis and autophagy in ovarian cancer via ROS overproduction	Upregulated LC3-II, Bax, caspase-3; downregulated Bcl-2
Maddah et al. [231]	AuNPs in HCT-116 colon cancer cells	Enhanced apoptosis via Bax, p53 upregulation, Bcl-2 downregulation	Significant chromatin condensation, apoptotic cell death
Radaic et al. [234]	Phosphatidylserine-capped AuNPs (PS-AuNPs)	Induced apoptosis in breast and prostate cancer cells	Increased caspase-3 activity, histone fragmentation
Zhang et al. [268]	AFt-C6 NPs	Dual-mode activation of apoptosis and autophagy in glioma cells; minimal cross-regulation	Caspase-3, LC3-II upregulation; downregulated p62
Pandey et al. [264]	AuNPs (~37 nm) in Lewis lung carcinoma (LLC) cells	Enhanced radiosensitization, DNA damage, therapeutic efficacy	Improved apoptotic, autophagic responses via ROS elevation
Xiaowei Ma et al. [246]	AuNPs in lysosomes	Disrupted lysosomal degradation; autophagosome accumulation	Increased LC3-II levels, p62 accumulation
Li et al. [252]	AuNPs in MRC-5 lung fibroblasts	Triggered autophagy, oxidative stress, lipid peroxidation	Increased MAP-LC3, ATG7, malondialdehyde (MDA) adducts
Ma et al. [261]	PEG-coated gold nanospikes (GNSs)	TAT-GNSs highest radiosensitization; enhanced ROS production, impaired autophagic flux	Increased LC3-II, p62 accumulation; enhanced apoptosis with 3-MA inhibitor

## Data Availability

This review article did not involve the generation or analysis of new data. As such, data sharing is not applicable. Nevertheless, any supporting materials or additional information relevant to the review may be provided by the corresponding author upon reasonable request.

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
