# Peer review of "Targeting Cancer Cell Fate: Apoptosis, Autophagy, and Gold Nanoparticles in Treatment Strategies"

_cimb, 2025, doi:10.3390/cimb47060460_

Round 1

Reviewer 1 Report

Comments and Suggestions for Authors

The work CIMB-3589163 presents an exhaustive review of the molecular interplay between apoptosis and autophagy in cancer, and how AuNPs could be used to modulate these pathways in order to improve cancer treatment. However, the manuscript is too long and tends to follow a repetitive format, which makes it somewhat overwhelming to read. Some sections should be revised to improve clarity and strengthen the overall presentation of the review.

  1. Mostly all the figures are taken from other works. It would be helpful to include a summary figure that visually integrates the most important aspects where AuNPs interact with apoptosis and autophagy. Additionally, the number of figures extracted from experimental papers should be reduced.
  2. In Figure 6, the authors mention: “as shown in Supporting Information Figures S2A and S2B” (lines 689–690). Is this referring to the original paper from which the figure was taken, or is there supplementary material associated with the current review? This is unclear and, in my opinion, this information may not be necessary.
  3. Figure 14 should be improved by adding labels and explaining the meaning of the initials or abbreviations used in the figure to enhance its readability.
  4. In line 176, when discussing the importance of apoptosis in maintaining tissue homeostasis, it may be valuable to briefly mention its role in tissue regeneration. This aspect is relevant to both homeostasis and cancer progression and could add depth to the discussion.
  5. As highlighted in the review, one of the main advantages of using AuNPs is their role as radiosensitizers to enhance the efficacy of radiotherapy. However, it would be beneficial to include a comparison with other types of nanoparticles that have also shown potential for radiosensitization and that may have reported interactions with apoptosis and autophagy. While I understand that the main focus is on AuNPs, presenting alternative nanoplatforms could enrich the review, especially considering that, as noted in the conclusions, AuNPs face limitations in their translational application.

Author Response

STATEMENT OF CORRECTIONS

REVISED MANUSCRIPT

Dear Reviewers,

In response to the Reviewers´ comments, the manuscript has been thoroughly revised. All corrections have been made in accordance with the Reviewers’ suggestions, and the changes are marked in the revised manuscript using yellow highlighting. Below follow the replies and respective answers to the Reviewers’ comments and enquires. The full set of answers has also been uploaded as a separate document. We sincerely thank you for your thoughtful feedback and efforts to improve the quality of our work.

Yours faithfully,

Dr. Maria Anthi Kouri

Reviewer 1

The work CIMB-3589163 presents an exhaustive review of the molecular interplay between apoptosis and autophagy in cancer, and how AuNPs could be used to modulate these pathways in order to improve cancer treatment. However, the manuscript is too long and tends to follow a repetitive format, which makes it somewhat overwhelming to read. Some sections should be revised to improve clarity and strengthen the overall presentation of the review.

Comment 1. Mostly all the figures are taken from other works. It would be helpful to include a summary figure that visually integrates the most important aspects where AuNPs interact with apoptosis and autophagy. Additionally, the number of figures extracted from experimental papers should be reduced.

Reply: Thank you for your helpful comments. We agree that the number of figures taken directly from experimental papers should be limited in a review, as they can interrupt the narrative and reduce conceptual clarity. Rather than attempting to summarize the complex and heterogeneous mechanisms of AuNP interactions with apoptosis and autophagy in a single schematic—which we found would be overly reductive—we have chosen to improve clarity by removing several figures (Figures 3, 6, and 10) that largely repeated experimental setups and added limited value to the conceptual flow.

In response to your suggestion, we have created a new Figure 2, which schematically illustrates the interactions between apoptosis and autophagy. We hope this addition enhances the manuscript's integrative value and supports readers in understanding these key cellular processes. While we did not include a similar figure for AuNPs due to the complexity involved, we believe the accompanying summary table will help convey the essential findings in a clear and accessible format.

Comment 2. In Figure 6, the authors mention: “as shown in Supporting Information Figures S2A and S2B” (lines 689–690). Is this referring to the original paper from which the figure was taken, or is there supplementary material associated with the current review? This is unclear and, in my opinion, this information may not be necessary.

Reply: We thank the reviewer for this thoughtful comment. The reference to “Supporting Information Figures S2A and S2B” was indeed from the original research article from which the figure was adapted. As this material is not part of the supplementary content for the current review and may cause confusion, we have removed the reference from the text as suggested.

Comment 3. Figure 14 should be improved by adding labels and explaining the meaning of the initials or abbreviations used in the figure to enhance its readability.

Reply: Thank you for this helpful suggestion. We have revised the figure legend to include the full forms of all abbreviations used (e.g., CT for chemotherapy, RT for radiotherapy), and ensured the description provides a clear understanding of each bar category. We believe this revision improves the readability and interpretability of the figure as recommended.

As can be depicted in the revised Manuscript:

Figure 11. Distribution of literature across key research domains involving gold nanoparticles (AuNPs) in cancer-related apoptosis and autophagy. This bar plot provides an indicative overview of studies published between 1970 and 2025 across selected thematic categories, including apop-tosis, autophagy, radiosensitization, drug delivery, chemotherapy (CT), and radiotherapy (RT), as well as their intersections with AuNP-based strategies. The plot also includes the distribution of in vitro, in vivo, and clinical investigations.

Comment 4. In line 176, when discussing the importance of apoptosis in maintaining tissue homeostasis, it may be valuable to briefly mention its role in tissue regeneration. This aspect is relevant to both homeostasis and cancer progression and could add depth to the discussion.

Reply: We thank the reviewer for this insightful suggestion. In response, we have revised the section titled "2.1.2. Role of Apoptosis in Maintaining Tissue Homeostasis" to include a concise but detailed discussion on the contribution of apoptosis to tissue regeneration. Drawing on recent literature, we now describe how apoptotic cells not only eliminate damaged tissue but also actively promote regeneration through the release of mitogenic signals (apoptosis-induced proliferation) and the resolution of inflammation via efferocytosis.

Additionally, we highlight the dual nature of these processes by noting their relevance to both regenerative healing and cancer progression. These additions are supported by recent experimental findings, including single-cell RNA-sequencing data in murine wound models and mechanistic studies on apoptotic signaling and stromal reprogramming. The updated text also incorporates relevant and up-to-date references from 2022–2024.

Revised Manuscript: (Lines 194-222)

Beyond its role in eliminating damaged or superfluous cells, apoptosis actively contributes to tissue regeneration. Dying cells can release mitogenic factors—such as Wnt proteins and prostaglandin E2 (PGE2)—which stimulate neighboring progenitor and stem cells to proliferate, a process known as apoptosis-induced proliferation (AiP) [52]. This has been demonstrated in several models, including regenerating skin, liver, and in Drosophila, where caspase activation in apoptotic cells drives compensatory proliferation [52].

In mammals, apoptotic cell clearance via efferocytosis further promotes regenera-tion by dampening inflammation and fostering a reparative microenvironment. Recent single-cell RNA sequencing in murine skin wounds revealed that apoptotic signaling and efferocytosis receptors, including Timd4 and Axl, are upregulated in fibroblasts and macrophages during early inflammation [53]. Functionally, Timd4 was shown to be essential for both apoptotic clearance and tissue repair, while Axl promoted regen-eration through separate, efferocytosis-independent pathways [53].

In chronic wounds such as diabetic ulcers, dysregulation of pathways like Axl/Gas6 impairs healing, highlighting the clinical relevance of apoptotic signaling in repair processes [53]. Across species—from Hydra to mammals—AiP is a conserved mechanism that mobilizes quiescent stem cells in response to injury, with caspase ac-tivity triggering regenerative proliferation [44,54–59].

In therapeutic settings, mesenchymal stromal cells (MSCs) administered for tissue repair often undergo apoptosis shortly after infusion. Their apoptotic clearance by monocytes reprograms immune responses toward anti-inflammatory, pro-regenerative phenotypes—supporting repair not by cell replacement, but by im-mune modulation [60].

However, the regenerative benefits of apoptotic signaling may be co-opted in cancer. Apoptosis induced by chemo- or radiotherapy can inadvertently stimulate sur-viving malignant cells through mitogenic cues, potentially facilitating tumor repopu-lation and therapy resistance [61]. Thus, apoptosis serves as both a regulator of tissue renewal and a potential driver of tumor progression, emphasizing the importance of context in its therapeutic targeting.

Comment 5. As highlighted in the review, one of the main advantages of using AuNPs is their role as radiosensitizers to enhance the efficacy of radiotherapy. However, it would be beneficial to include a comparison with other types of nanoparticles that have also shown potential for radiosensitization and that may have reported interactions with apoptosis and autophagy. While I understand that the main focus is on AuNPs, presenting alternative nanoplatforms could enrich the review, especially considering that, as noted in the conclusions, AuNPs face limitations in their translational application.

Reply: We thank the reviewer for this thoughtful suggestion. In response, we have added a brief paragraph following Section 3.1.1 to acknowledge and present recent examples of alternative nanoparticle platforms—such as titanium dioxide, iron oxide, and hafnium oxide nanoparticles—that have shown potential as radiosensitizers through their interactions with apoptosis and autophagy pathways.

While we chose not to expand this section extensively due to the already broad scope of the review, we felt it was important to familiarize readers with the fact that other nanomaterials are also being explored in this context. This addition adds perspective to the discussion, particularly in light of the translational challenges associated with AuNPs.

Revised Manuscript: (Lines 478-493)

Of course, it is important to acknowledge that gold nanoparticles are not the only nanomaterials with radiosensitizing capabilities. Several other platforms have shown promise in enhancing radiotherapy through modulation of apoptosis and autophagy. For instance, titanium dioxide (TiO₂) nanoparticles can generate reactive oxygen spe-cies under ionizing or UV-A radiation, triggering oxidative stress and apoptotic cell death [168]. Similarly, iron oxide nanoparticles (IONPs) have been found to enhance radiosensitivity by promoting ferroptosis and autophagy-related mechanisms, partic-ularly in challenging tumor microenvironments [169–172]. A clinically advanced ex-ample is hafnium oxide (HfO₂) nanoparticles, such as NBTXR3, which increase radia-tion dose deposition and induce apoptosis in solid tumors through their high atomic number and interaction with therapeutic radiation [169,171,173]. That said, gold na-noparticles still offer distinct advantages that support their continued prominence in this field. Their high degree of surface tunability, well-documented biocompatibility, and dual ability to modulate apoptotic and autophagic pathways make them uniquely versatile. While other nanomaterials are promising in specific contexts, AuNPs remain among the most adaptable and extensively studied radiosensitizers with translational potential [150].

Reviewer 2 Report

Comments and Suggestions for Authors

In this review, the authors did comprehensive analysis about how gold nanoparticles as radiosensitizers and as drug carriers modulate apoptosis and autophagy in cancer treatment and also described the mechanisms of action of gold nanoparticles. This review is important as it brings together research papers regarding the interaction of gold nanoparticles and the processes of apoptosis and autophagy in cancer treatment. This is a nicely constructed and mostly error-free review, I have some comments the authors should consider:

Line 239; Authors wrote that different cellular stressors, such as nutrient deprivation and metabolic stress, can trigger autophagy activation through the overexpression of the enzyme AMPK. AMPK is not activated by overexpression it is generally activated following energy stress by phosphorylation of liver kinase B1 (LKB1) and calcium/calmodulin-dependent protein kinase kinase 2 (CAMKK2) (10.1016/j.molmet.2024.102042).

Line 242; Authors wrote that the ULK1 complex is essential for stimulating beclin-1 expression, which further initiates the formation of the VPS34 complex. The main mechanism which ULK1 stimulates VPS34 complex is by phosphorylation of beclin-1 at Ser15 and Ser30 and ATG14 at Ser29 (doi:10.1038/ncb2757, doi.org/10.1080/15548627.2017.1422851, doi:10.1080/15548627.2016.1140293,). The authors need to change that in the text and Figure 1. As Figure 1 is licensed authors need to draw or find appropriate picture that illustrate this step of macroautophagy. In addition, authors wrote that autophagosomes encapsulate damaged organelles and misfolded proteins, but in Figure 1 only mitochondria is within autophagosome which could represent selective autophagy of mitochondria (mitophagy). I suggest authors to draw or find an appropriate picture for Figure 1.   

Line 258; Authors describe autophagy as the encapsulation of cytoplasmic components into double or multilayer vacuoles to form autophagosomes. I would avoid describing autophagosomes as multilayer vacuoles because autophagosomes typically have a double membrane. According to Guidelines for the use and interpretation of assays for monitoring autophagy (4th edition) (10.1080/15548627.2020.1797280) it is possible to observe multi-lamellar membrane structures which may be multiple double layers of phagophores, autolysosomes or just artifact of fixation.

Figure 2 is confusing. In the picture, it is not clear what ATG3 (-) represents. I found in the text that caspase cleaves ATG3 which inhibits autophagosome formation, but that is not clearly represented in the picture. Furthermore, in the picture, it looks like that arrow shows that autophagy-mediated degradation of caspase-8 stimulates apoptosis instead of suppressing it. Authors need to draw new figure or find appropriate one. It also needs to mention Figure 2 in the text where is necessary.

In the section "Autophagy enhancement by AuNPs-Based Drug Delivery" authors wrote that delivery systems involving AuNPs loaded with small interfering RNA (siRNA), short-hairpin RNA (shRNA), or CRISPR/Cas9 which target ATG12-ATG5-ATG16L1 can improve therapeutic outcomes by suppressing tumor cell resistance and enhancing autophagic responses. Using siRNA/shRNA which targets ATG12, ATG5 or ATG16L1 autophagy will be supress, not induce. I would omit these genetic tools for autophagy inhibition from this section as it is about autophagy enhancement by AuNPs-based drug delivery.  Furthermore, in this section authors wrote that AuNPs can alter DNA methylation, histone acetylation, and miRNA expression patterns, significantly affecting autophagy-related genes such as Beclin-1, ATG5, and LC3. It is important to clarify whether AuNPs upregulate or downregulate these autophagy-related genes as this section is about autophagy enhancement by AuNPs-based drug delivery.

In the section "Regulation of Apoptosis on Autophagy" there are some author-date citations that should be removed as author number references are used.

In Figure 14 authors need to write in the figure legend the full name of abbreviations for CT and RT.

Author Response

STATEMENT OF CORRECTIONS

REVISED MANUSCRIPT

Dear Reviewers,

In response to the Reviewers´ comments, the manuscript has been thoroughly revised. All corrections have been made in accordance with the Reviewers’ suggestions, and the changes are marked in the revised manuscript using yellow highlighting. Below follow the replies and respective answers to the Reviewers’ comments and enquires. The full set of answers has also been uploaded as a separate document. We sincerely thank you for your thoughtful feedback and efforts to improve the quality of our work.

Yours faithfully,

Dr. Maria Anthi Kouri

Reviewer 2

In this review, the authors did comprehensive analysis about how gold nanoparticles as radiosensitizers and as drug carriers modulate apoptosis and autophagy in cancer treatment and also described the mechanisms of action of gold nanoparticles. This review is important as it brings together research papers regarding the interaction of gold nanoparticles and the processes of apoptosis and autophagy in cancer treatment. This is a nicely constructed and mostly error-free review, I have some comments the authors should consider:

Comment 1. Line 239; Authors wrote that different cellular stressors, such as nutrient deprivation and metabolic stress, can trigger autophagy activation through the overexpression of the enzyme AMPK. AMPK is not activated by overexpression it is generally activated following energy stress by phosphorylation of liver kinase B1 (LKB1) and calcium/calmodulin-dependent protein kinase kinase 2 (CAMKK2) (10.1016/j.molmet.2024.102042).

Reply: Thank you for the valuable correction and for providing a relevant and recent reference. We agree with your observation and thus we have revised the text accordingly to accurately reflect the activation mechanisms of AMPK. We have also included the suggested reference. As can be depicted in the revised manuscript:

 (Line 268):

" Different cellular stressors, such as nutrient deprivation and metabolic stress, can trigger autophagy activation through the phosphorylation and subsequent activation of 5′-AMP-activated protein kinase (AMPK), primarily mediated by upstream kinases such as liver kinase B1 (LKB1) and calcium/calmodulin-dependent protein kinase kinase 2 (CAMKK2) [78–80].

Comment 2. Line 242; Authors wrote that the ULK1 complex is essential for stimulating beclin-1 expression, which further initiates the formation of the VPS34 complex. The main mechanism which ULK1 stimulates VPS34 complex is by phosphorylation of beclin-1 at Ser15 and Ser30 and ATG14 at Ser29 (doi:10.1038/ncb2757, doi.org/10.1080/15548627.2017.1422851, doi:10.1080/15548627.2016.1140293. The authors need to change that in the text and Figure 1. As Figure 1 is licensed authors need to draw or find appropriate picture that illustrate this step of macroautophagy. In addition, authors wrote that autophagosomes encapsulate damaged organelles and misfolded proteins, but in Figure 1 only mitochondria is within autophagosome which could represent selective autophagy of mitochondria (mitophagy). I suggest authors to draw or find an appropriate picture for Figure 1.   

Reply: We thank the reviewer for this helpful and detailed comment. In response, we have revised the Manuscript to more precisely describe the accurate mechanism:

(Lines 272-276):

“Activated AMPK, in turn, promotes autophagy by inhibiting the mTOR pathway and activating the ULK1 complex, which is composed of ULK1, FIP200, ATG13, and ATG101. The ULK1 complex plays a crucial role in autophagy initiation by phosphorylating Beclin-1 at Ser15 and Ser30, as well as ATG14 at Ser29, particularly under conditions of energy stress or mTORC1 [81–83].”

The updated section incorporates the references suggested by the Reviewer (Russell et al., 2013; Park et al., 2018; Park et al., 2016), which we have also added to the reference list.

Regarding Figure 1, we acknowledge the Reviewer’s observation that it specifically depicts the selective degradation of a mitochondrion (mitophagy), while the accompanying text previously referred more broadly to autophagosomal degradation of various cargo types. To address this, we have revised the relevant sentence to explicitly state that Figure 1 illustrates mitophagy — a specific form of autophagy involving the degradation of damaged mitochondria. The revised sentence now reads (Lines 248-255):

“The subsequent formation of autophagosomes involves the lipidation of LC3-I (light chain 3-I) to produce LC3-II, a process that requires the action of proteins such as ATG5 and ATG16L1 [75–77]. Following the formation of the autophagosome, it fuses with a ly-sosome, a process facilitated by proteins such as LAMP2 and SNAREs [77]. This fusion allows the degradation of the autophagosomal cargo, which involves various enzymes, including proteases, lipases, and nucleases [77]. “As illustrated in Figure 1, this process is exemplified by the selective degradation of a damaged mitochondrion, representing mitophagy as a specific form of autophagy [77].”

Comment 3. Line 258; Authors describe autophagy as the encapsulation of cytoplasmic components into double or multilayer vacuoles to form autophagosomes. I would avoid describing autophagosomes as multilayer vacuoles because autophagosomes typically have a double membrane. According to Guidelines for the use and interpretation of assays for monitoring autophagy (4th edition) (10.1080/15548627.2020.1797280) it is possible to observe multi-lamellar membrane structures which may be multiple double layers of phagophores, autolysosomes or just artifact of fixation.

Reply: Thank you for the close attention to detail and the valuable clarification you provided. In accordance with the Guidelines for the use and interpretation of assays for monitoring autophagy (4th edition) by Klionsky et al. and your suggestion, we have revised the sentence to describe autophagosomes as double-membraned vesicles and removed the reference to multilayer vacuoles. We have also added a clarifying note, as recommended in the guidelines, that multilamellar structures may be observed but are not typical of canonical autophagosomes.

Revised Manuscript (Line 288-297):

“Autophagy, which is often referred to as self-digestion, is a well-conserved process that plays an essential role in maintaining cellular health, differentiation, and overall homeostasis [58,78,79]. It typically involves two stages: the encapsulation of cytoplasmic components into double-membraned vesicles known as autophagosomes, followed by their fusion with lysosomes to form autolysosomes, where degradation occurs [61,80]. According to the Guidelines for the use and interpretation of assays for monitoring autophagy (4th edition) by Klionsky et al., autophagosomes are typically double-membraned; although multilamellar membrane structures can occasionally be observed, these are not characteristic and may reflect overlapping phagophores, autolysosomes, or artifacts of sample preparation [87].”

Comment 4. Figure 2 is confusing. In the picture, it is not clear what ATG3 (-) represents. I found in the text that caspase cleaves ATG3 which inhibits autophagosome formation, but that is not clearly represented in the picture. Furthermore, in the picture, it looks like that arrow shows that autophagy-mediated degradation of caspase-8 stimulates apoptosis instead of suppressing it. Authors need to draw new figure or find appropriate one. It also needs to mention Figure 2 in the text where is necessary.

Reply: Thank you for your helpful and insightful comment. In response, we have replaced the original Figure 2 with a newly designed schematic that clearly illustrates the key regulatory mechanisms involved in the crosstalk between autophagy and apoptosis.

  • The caspase-mediated cleavage of ATG3, which inhibits autophagy,
  • The autophagy-mediated degradation of caspase-8, which suppresses apoptosis, and
  • The role of autophagic clearance in maintaining cellular homeostasis and preventing stress-induced apoptosis.

We also addressed the concern about potential confusion by relocating the figure reference within the text. Specifically, Figure 2 is now mentioned at the end of Section 2.3.2, after both 2.3.1 and 2.3.2 describe the bidirectional regulation between autophagy and apoptosis in detail. This positioning allows the figure to serve as a clear, visual summary of the mechanisms discussed, without interrupting the flow of explanation or introducing confusion.

The changes are depicted in the Revised Manuscript:

(Lines 324-331):

“As has been previously described, autophagy and apoptosis constitute two fundamental processes governing cell fate, both of which are intricately linked in cancer progression and therapeutic response. Their interaction exhibits a dual role, with autophagy either facilitating apoptotic cell death or promoting tumor cell survival, depending on the cellular context [43]. As is illustrated in Figure 2, these pathways are closely interconnected through both direct molecular interactions and context-dependent signaling feedback loops, with each process capable of regulating or suppressing the other under specific physiological or pathological conditions.”

(Lines 366-367):

“A summary of the aforementioned key interactions between autophagy and apoptosis is illustrated in Figure 2.”

(Lines 369-374):

Figure 2. Schematic illustration of the crosstalk between autophagy and apoptosis. This diagram highlights the core regulatory mechanisms linking autophagy and apoptosis. Caspases, central to apoptotic execution, inhibit autophagy by cleaving key proteins such as ATG3. Autophagy, in turn, can suppress apoptosis by degrading pro-apoptotic factors like caspase-8 and maintaining cellular homeostasis through the clearance of damaged organelles and protein aggregates. Impaired or excessive autophagy may instead promote apoptosis.

Comment 5. In the section "Autophagy enhancement by AuNPs-Based Drug Delivery" authors wrote that delivery systems involving AuNPs loaded with small interfering RNA (siRNA), short-hairpin RNA (shRNA), or CRISPR/Cas9 which target ATG12-ATG5-ATG16L1 can improve therapeutic outcomes by suppressing tumor cell resistance and enhancing autophagic responses. Using siRNA/shRNA which targets ATG12, ATG5 or ATG16L1 autophagy will be suppress, not induce. I would omit these genetic tools for autophagy inhibition from this section as it is about autophagy enhancement by AuNPs-based drug delivery.  Furthermore, in this section authors wrote that AuNPs can alter DNA methylation, histone acetylation, and miRNA expression patterns, significantly affecting autophagy-related genes such as Beclin-1, ATG5, and LC3. It is important to clarify whether AuNPs upregulate or downregulate these autophagy-related genes as this section is about autophagy enhancement by AuNPs-based drug delivery.

Reply: Thank you for your comment! Regarding the first point, we agree that delivery systems involving siRNA, shRNA, or CRISPR/Cas9 targeting ATG12, ATG5, or ATG16L1 result in autophagy inhibition rather than enhancement. As this was inconsistent with the focus of the section, we have revised the text to clarify this distinction and removed the suggestion that these tools enhance autophagy.

(Lines 875-884):

“Expanding on nanoparticle functionality, researchers have explored the use of AuNPs to improve the delivery of genetic tools in cancer therapy [255,256]. While some systems involve AuNP-mediated delivery of small interfering RNA (siRNA), short-hairpin RNA (shRNA), or CRISPR/Cas9 targeting autophagy-related genes such as ATG12, ATG5, or ATG16L1, these approaches are primarily used to inhibit autophagy, particularly in cases where excessive autophagy contributes to tumor survival. As such, they are not discussed in detail here, as this section focuses on strategies aimed at enhancing autophagy. Nonetheless, the ability of AuNPs to facilitate gene delivery remains a promising therapeutic platform, especially in overcoming intracellular delivery barriers and improving treatment precision [77,257–259].”

Regarding the second point, we appreciate the reviewer’s observation and have conducted further background research. Multiple studies demonstrate that the effect of AuNPs on autophagy-related genes such as Beclin-1, ATG5, and LC3 is highly context-dependent, influenced by factors such as nanoparticle size, surface properties, coating agents, and cellular environment. In many cancer models, AuNPs have been shown to upregulate these markers and enhance autophagic activity. However, certain formulations may impair lysosomal function and suppress autophagic flux. We have revised the text accordingly to reflect this dual role and supported the update with appropriate citations to clarify the underlying mechanisms and enhance the accuracy of the section.

(Lines 885-902):

“Recent research emphasizes that AuNPs influence epigenetic and molecular mechanisms critical for autophagy modulation and apoptosis in cancer treatment [260,261]. AuNPs can alter DNA methylation, histone acetylation, and miRNA expres-sion patterns, significantly affecting autophagy-related genes such as Beclin-1, ATG5, and LC3 [262,263]. Several studies have reported that AuNPs upregulate these au-tophagy markers and enhance autophagic activity. For instance, Zhang et al. demon-strated that PEG-AuNPs increased LC3 and Beclin-1 levels while modulating au-tophagic flux in tumor-associated macrophages [268]. Similarly, Zhang et al. (2021) described nanoparticle-induced upregulation of LC3-II and Beclin-1 as central to au-tophagy induction in breast cancer cells [86]. These findings suggest that properly de-signed AuNPs may enhance autophagy through transcriptional or post-translational regulation of key proteins. However, the effects of AuNPs on autophagy are con-text-dependent. Certain formulations can impair lysosomal function, leading to au-tophagosome accumulation and disrupted autophagic flux, as shown by Ma et al. and Siddiqi et al. [249,259] . Such outcomes emphasize the importance of nanoparticle size, coating, and cellular context in determining biological effects. Overall, AuNP-mediated modulation of autophagy—when properly engineered—offers a promising strategy to enhance chemotherapeutic efficacy and target drug-resistant cancer cells [170].”

Comment 6. In the section "Regulation of Apoptosis on Autophagy" there are some author-date citations that should be removed as author number references are used.

Reply: Thank you for pointing this out. We have carefully reviewed the section and removed the author-date citations to ensure consistency.

Comment 7. In Figure 14 authors need to write in the figure legend the full name of abbreviations for CT and RT.

Reply: Thank you for the helpful suggestion. We have revised the legend of Figure 14 to include the full names chemotherapy (CT) and radiotherapy (RT) within the text for clarity.

Reviewer 3 Report

Comments and Suggestions for Authors

This manuscript suffer from severe pitfalls:

-Lack of Conceptual Cohesion and Coherence

The manuscript attempts to link three major topics—apoptosis, autophagy, and gold nanoparticles (AuNPs)—into a unified narrative. However, it often feels like a juxtaposition of three distinct reviews rather than an integrated discussion. The interplay between apoptosis and autophagy is well-established, but the introduction of AuNPs feels more like an appendage than a mechanistically cohesive part of the narrative.

 The manuscript would benefit from splitting this into two separate reviews: one on the interplay of apoptosis and autophagy in cancer (but this topic is not new and so many reviews have been already published), and another on how AuNPs interact with these pathways. The latter can be also part of one review but differently written better integrating the three topics.

Overload of Detail with Limited Synthesis

While comprehensive, the text is densely packed with experimental details and literature citations, often without offering synthesis, critique, or hierarchy of findings.

Example: Pages are devoted to listing studies using AuNPs in radiosensitisation and drug delivery, but with minimal commentary on:

Comparative efficacy

Mechanistic commonalities

Contradictions

Translation to clinical settings

Language and Style Issues

The manuscript contains numerous stylistic issues that highly affect clarity:

Repetitive use of rhetorical phrases (e.g., “this dual role,” “this interplay,” “a paradigm shift”) without grounding them in specific novel insight. Obscure, verbose language in parts of the abstract and introduction. Redundant paragraphs, especially those reiterating basic definitions of apoptosis and autophagy.

Ambiguity in Translational Relevance

The abstract and conclusion mention "patient-centered precision therapies" and "paradigm shifts," but there is little actual translational insight offered. The current manuscript reads as preclinical and descriptive, with no real pathway to clinical application.

The paper needs to add a "Clinical Perspective" subsection to each major section or as a final discussion, outlining: Known clinical trials (if any); Pharmacokinetics/toxicity issues with AuNPs; Regulatory challenges; Patient stratification criteria (e.g., biomarkers)

Figures and Legends

Some of the figures are poorly integrated:

Many seem borrowed from other publications and lack proper visual coherence within the manuscript. Their placement often interrupts the text rather than supporting it. Several figures (e.g., Figures 3, 4, 5) depict microscopy images or survival curves from other studies, which is problematic in a review—especially when over-interpreted.

It is needed in a future review to replace them with schematic overviews or original conceptual diagrams explaining the tripartite interaction among apoptosis, autophagy, and AuNPs.

Other Concerns

Many references are outdated or repetitive. Consolidate where possible.

Terms like “triptych” (line 120) are more poetic than helpful in a scientific review.

The role of ROS is described redundantly across different sections.

Several citation styles are inconsistent (e.g., “[97],” “(Nishida et al., 2009),” etc.).

Comments on the Quality of English Language

Lot of language and style Issues

The manuscript contains numerous stylistic issues that highly affect clarity:

Repetitive use of rhetorical phrases (e.g., “this dual role,” “this interplay,” “a paradigm shift”) without grounding them in specific novel insight. Obscure, verbose language in parts of the abstract and introduction. Redundant paragraphs, especially those reiterating basic definitions of apoptosis and autophagy.

Author Response

STATEMENT OF CORRECTIONS

REVISED MANUSCRIPT

Dear Reviewers,

In response to the Reviewers´ comments, the manuscript has been thoroughly revised. All corrections have been made in accordance with the Reviewers’ suggestions, and the changes are marked in the revised manuscript using yellow highlighting. Below follow the replies and respective answers to the Reviewers’ comments and enquires. The full set of answers has also been uploaded as a separate document. We sincerely thank you for your thoughtful feedback and efforts to improve the quality of our work.

Yours faithfully,

Dr. Maria Anthi Kouri

Reviewer 3

Comment 1. -Lack of Conceptual Cohesion and Coherence

The manuscript attempts to link three major topics—apoptosis, autophagy, and gold nanoparticles (AuNPs)—into a unified narrative. However, it often feels like a juxtaposition of three distinct reviews rather than an integrated discussion. The interplay between apoptosis and autophagy is well-established, but the introduction of AuNPs feels more like an appendage than a mechanistically cohesive part of the narrative.

The manuscript would benefit from splitting this into two separate reviews: one on the interplay of apoptosis and autophagy in cancer (but this topic is not new and so many reviews have been already published), and another on how AuNPs interact with these pathways. The latter can be also part of one review but differently written better integrating the three topics.

Reply: We sincerely thank the Reviewer for the comments and critical observations. We understand the perspective regarding the perceived compartmentalization of apoptosis, autophagy, and gold nanoparticles within our review. However, the fundamental goal of our manuscript is to bridge these topics in a unified narrative. Specifically, we aimed to elucidate the complex interplay between apoptosis and autophagy mechanisms and explore how AuNPs potentially modulate each mechanism and their interplay within the context of cancer treatment.

While apoptosis and autophagy pathways in cancer is indeed documented, the novelty and primary focus of our review lie in investigating how AuNPs influence these cellular pathways. More specifically, we explore the potential impacts of AuNPs in therapeutic strategies, including radiosensitization and drug delivery, and how they may modulate apoptosis, autophagy and their crosstalk. The detailed presentation of research after research within our manuscript is intentional and highlights the fact that this subject is still not fully explored or understood. Our aim was to comprehensively present existing data, identify current research gaps, and encourage further investigation into these crucial mechanisms and interactions.

Splitting the manuscript into separate reviews, as kindly suggested, would dilute this core objective and lose the integrative perspective we aimed to provide. Instead, we have carefully restructured sections, improved thematic continuity, and introduced new figures to clarify and reinforce this integrated approach. Our intent is to present a cohesive and mechanistically grounded discussion that can serve as a reference point for future studies aiming at better understanding and optimizing AuNP-based therapies in cancer.

We deeply appreciate the constructive critique provided by Reviewer 3, which has allowed us to clarify and enhance the integrative rationale of our manuscript.

Comment 2. Overload of Detail with Limited Synthesis

While comprehensive, the text is densely packed with experimental details and literature citations, often without offering synthesis, critique, or hierarchy of findings.

Example: Pages are devoted to listing studies using AuNPs in radiosensitisation and drug delivery, but with minimal commentary on:

Comparative efficacy

Mechanistic commonalities

Contradictions

Translation to clinical settings.   

Reply: Thank you for your insightful feedback. We  appreciate the importance of presenting comparative efficacy, mechanistic commonalities, contradictions, and the translational aspects of gold nanoparticle in cancer treatment. These are indeed critical elements, and we share your view on their relevance. We have attempted to address each of these points—comparative and mechanistic analysis, identification of conflicting findings, and implications for clinical translation—throughout the main discussion (Section 5), particularly within the structured sub-sections as well as in the dedicated "Limitations" and "Future Directions" sections.

To support this narrative, we have included Table 1, which summarizes the key findings, mechanisms, and biological responses reported across a broad spectrum of AuNP studies. Additionally, Figure 14 provides an indicative overview of current research trends, highlighting both the focus areas and existing gaps, thereby helping to contextualize our synthesis and underscore areas needing further exploration.

We fully agree with your observation regarding the scarcity of clinical applications. Indeed, our own literature review confirmed that clinical studies involving AuNPs remain extremely limited. In light of your comment, we have added a dedicated paragraph (Section 5.2.1. Clinical Translation and Regulatory Considerations) that expands on this issue and is also addressing the translational gap highlighted in Comment 4.

Comment 3. Language and Style Issues

The manuscript contains numerous stylistic issues that highly affect clarity:

Repetitive use of rhetorical phrases (e.g., “this dual role,” “this interplay,” “a paradigm shift”) without grounding them in specific novel insight. Obscure, verbose language in parts of the abstract and introduction. Redundant paragraphs, especially those reiterating basic definitions of apoptosis and autophagy.

Reply: We appreciate the reviewer’s feedback regarding stylistic clarity. In response, we have revised the manuscript to make the language more concise, scientific, and precise. We have eliminated redundant rhetorical phrases and attempted to enhance clarity and focus.

Regarding the basic definitions of apoptosis and autophagy, we have altered these sections to avoid redundancy while retaining the essential concepts and also following the other reviewers comments. This ensures accessibility for readers from a broad range of scientific backgrounds. Given that our review spans multiple disciplines—including physics, medical physics, biology, radiobiology, chemistry, medicine, and oncology—we believe it is important to serve as a connective tissue across this multidisciplinary domain. Therefore, we aimed to maintain a balance between scientific rigor and conceptual clarity to foster a common understanding among diverse readers in the context of cancer research.

Comment 4. Ambiguity in Translational Relevance

The abstract and conclusion mention "patient-centered precision therapies" and "paradigm shifts," but there is little actual translational insight offered. The current manuscript reads as preclinical and descriptive, with no real pathway to clinical application.

The paper needs to add a "Clinical Perspective" subsection to each major section or as a final discussion, outlining: Known clinical trials (if any); Pharmacokinetics/toxicity issues with AuNPs; Regulatory challenges; Patient stratification criteria (e.g., biomarkers)

Reply: We thank the reviewer for this valuable observation. In response, we have added a new subsection (5.2.1. Clinical Translation and Regulatory Considerations) that specifically addresses the translational landscape. This section summarizes known clinical trials (e.g., NCT01270139 and the AuroShells study in prostate cancer), discusses pharmacokinetics and toxicity concerns related to organ retention, and highlights regulatory challenges due to the lack of harmonized evaluation standards.

We now hope to link the preclinical modulation of apoptosis and autophagy by AuNPs to the translational gap, emphasizing the absence of clinical studies investigating these mechanisms directly. We also address patient stratification, referencing tumor heterogeneity and the role of predictive biomarkers and imaging technologies (e.g., radiomics) in advancing precision nanomedicine.

Revised Manuscript (Lines 1162-1196)

5.2.1. Clinical Translation and Regulatory Considerations

While numerous preclinical studies investigating AuNPs in the context of apopto-sis and autophagy have demonstrated promising results, clinical translation remains minimal. To date, the clinical implementation of AuNP-based therapies has been ex-tremely limited [9]. A notable example includes a first-in-human clinical trial (NCT01270139) evaluating the safety and feasibility of gold nanoshell-mediated pho-tothermal ablation, which reported a favorable safety profile and represented an initial proof-of-concept in humans [306]. Similarly, AuroShells—commercially developed gold nanoparticles—have been tested in a small-scale clinical study for prostate cancer treatment, demonstrating technical feasibility and localized control. Still though, sig-nificant translational challenges persist. As detailed in the previous section, biodistri-bution and long-term organ retention remain key concerns for safety, particularly re-garding accumulation in the liver, spleen, and kidneys [307,308]. While certain formu-lations, such as glutathione-coated ultrasmall AuNPs, have shown improved clearance and reduced toxicity, comprehensive pharmacokinetic profiling across diverse nano-particle designs is still lacking [290].

Crucially, while many preclinical studies have demonstrated the capacity of AuNPs to modulate key cell death pathways—namely, apoptosis and autophagy—the clinical implications of these effects remain largely unexplored [39]. The absence of clinical trials directly evaluating these mechanistic pathways underscores a significant translational gap. Bridging this gap will require not only efficacy and safety validation but also mechanistic biomarkers that can link AuNP exposure to pathway-specific therapeutic outcomes in human tumors [309,310].

On the regulatory front, the lack of harmonized standards for evaluating the effi-cacy and safety of nanomedicine products complicates approval [311]. AuNPs exhibit unique interactions at the molecular and cellular levels, which traditional drug evalu-ation frameworks may not fully capture [9]. As such, the development of tailored reg-ulatory guidelines is essential to advance clinical adoption.

Another central issue is patient selection. Tumor heterogeneity significantly in-fluences AuNP uptake and the expression of apoptotic and autophagic machinery [312]. This makes stratification strategies critical for future trials. Recent studies un-derscore the value of predictive biomarkers and non-invasive imaging tools—such as radiomics and functional imaging—to estimate AuNP accumulation and identify tu-mors likely to respond based on their apoptotic or autophagic profiles [313,314]. These techniques hold potential for enabling precision nanomedicine by identifying patients most likely to benefit from AuNP-based interventions targeting these pathways.”

Comment 5. Figures and Legends

Some of the figures are poorly integrated:

Many seem borrowed from other publications and lack proper visual coherence within the manuscript. Their placement often interrupts the text rather than supporting it. Several figures (e.g., Figures 3, 4, 5) depict microscopy images or survival curves from other studies, which is problematic in a review—especially when over-interpreted.

It is needed in a future review to replace them with schematic overviews or original conceptual diagrams explaining the tripartite interaction among apoptosis, autophagy, and AuNPs.

Reply: We appreciate the reviewer’s observation regarding the visual coherence and relevance of the figures. We fully agree that in a review context, figures should serve to clarify and integrate concepts rather than disrupt the flow or reiterate data from primary research.

In response, we have removed figures (3,6,10) that were either confusing, visually inconsistent, or merely repeated microscopy images or survival curves from other studies without sufficient added value. We have also replaced Figure 2 with an original schematic overview illustrating the interaction between apoptosis and autophagy, to provide conceptual clarity and improve continuity with the surrounding text.

Regarding the involvement of AuNPs, we acknowledge the complexity and heterogeneity of the underlying mechanisms, which made it challenging to distill into a single coherent schematic. Instead, we chose to retain and refine the summary table, which we believe more effectively synthesizes the key findings and provides a clear, comparative view of AuNP implications across different studies.

Comment 6. Other Concerns

Many references are outdated or repetitive. Consolidate where possible.

Terms like “triptych” (line 120) are more poetic than helpful in a scientific review.

The role of ROS is described redundantly across different sections.

Several citation styles are inconsistent (e.g., “[97],” “(Nishida et al., 2009),” etc.).

Reply: We thank the reviewer for the constructive observations. We have carefully revised the reference list to remove outdated or redundant sources and have updated the manuscript with more recent and relevant citations where appropriate. Citation formatting has been standardized throughout the manuscript for consistency.

Regarding the role of reactive oxygen species (ROS), we have examined the relevant sections and ensured that their discussion is not redundant but contextually distinct—highlighting ROS in relation to specific mechanistic pathways such as apoptosis, autophagy, and radiosensitization as they arise in different therapeutic contexts. We hope these revisions improve the clarity and cohesion of the manuscript.

Comment 7. Comments on the Quality of English Language

Lot of language and style Issues

The manuscript contains numerous stylistic issues that highly affect clarity:

Repetitive use of rhetorical phrases (e.g., “this dual role,” “this interplay,” “a paradigm shift”) without grounding them in specific novel insight. Obscure, verbose language in parts of the abstract and introduction. Redundant paragraphs, especially those reiterating basic definitions of apoptosis and autophagy

Reply: Thank you for your comment. These issues have been addressed as noted in our response to Comment 3. We have revised the language for clarity, removed redundant phrases, and condensed background sections while retaining key concepts for readers from various disciplines.

Round 2

Reviewer 3 Report

Comments and Suggestions for Authors

In my opinion on this manuscript remains very very critical being the manuscript stil plenty of critical parts and the amendments made by the Authors are largely insufficient.

What they wrote and amended in the text are a mere lining of the study, albeit the manuscript is ameliorated but not enough.

In addition the manuscript has a number excessive of citations most of them totally not necessary. Maybe a half of them must be accepted.

Finally true final Perspectives are needed.

A note: the Author's list at the end has an "and" then nothing.

Comments on the Quality of English Language

Still several problems of editing.

Author Response

STATEMENT OF CORRECTIONS

REVISED MANUSCRIPT- Round 2

Dear Reviewers,

In response to the Reviewers´ comments, the manuscript has been thoroughly revised. All corrections have been made in accordance with the Reviewers’ suggestions, and the changes are marked in the revised manuscript using yellow highlighting. Below follow the replies and respective answers to the Reviewers’ comments and enquires. The full set of answers has also been uploaded as a separate document. We sincerely thank you for your thoughtful feedback and efforts to improve the quality of our work.

Yours faithfully,

Dr. Maria Anthi Kouri

Reviewer 3

In my opinion on this manuscript remains very very critical being the manuscript stil plenty of critical parts and the amendments made by the Authors are largely insufficient

What they wrote and amended in the text are a mere lining of the study, albeit the manuscript is ameliorated but not enough.

In addition the manuscript has a number excessive of citations most of them totally not necessary. Maybe a half of them must be accepted.

Finally true final Perspectives are needed.

A note: the Author's list at the end has an "and" then nothing.

Reply: We thank the Reviewer for taking the time to review our revised manuscript and for their continued feedback.

On the reviewer's opinion that the amendments are still insufficient, we would like to respectfully note that substantial revisions were made in response to the previous round of comments. These included structural improvements, clarification of key concepts, and expanded discussion in several sections to enhance the overall quality and clarity of the manuscript following the suggestions of all 3 Reviewers.

Regarding the number of references, we would like to clarify that the citations included are all essential to support the scientific content of the paper. Many of these references were added specifically to address previous comments raised by Reviewers 1 and 2, who requested a more thorough contextualization of our findings. We believe that reducing the number of citations significantly, as suggested, would risk undermining the completeness and scientific rigor of the discussion.

Lastly, we thank the Reviewer for pointing out the formatting issue in the author list. This has now been corrected.

Once again, we appreciate the Reviewer’s feedback and hope that the revised manuscript now meets the standards for publication.

Round 3

Reviewer 3 Report

Comments and Suggestions for Authors

Albeit the Authors forgot to add the new perspectives the manuscript is a bit better.